# Bridging of host-microbiota tryptophan partitioning by the serotonin pathway in fungal pneumonia

Giorgia Renga[1], Fiorella D'Onofrio[1], Marilena Pariano[1], Roberta Galarini[2], Carolina Barola[2], Claudia Stincardini[1], Marina M. Bellet[1], Helmut Ellemunter[3], Cornelia Lass-Flörl[4], Claudio Costantini[1], Valerio Napolioni [5], Allison K. Ehrlich[6], Cinzia Antognelli[1], Massimo Fini[7], Enrico Garaci[7], Emilia Nunzi[1,8] & Luigina Romani [1,7,8] ✉

The aromatic amino acid L-tryptophan (Trp) is essentially metabolized along the host and microbial pathways. While much is known about the role played by downstream metabolites of each pathways in intestinal homeostasis, their role in lung immune homeostasis is underappreciated. Here we have examined the role played by the Trp hydroxylase/5-hydroxytryptamine (5-HT) pathway in calibrating host and microbial Trp metabolism during *Aspergillus fumigatus* pneumonia. We found that 5-HT produced by mast cells essentially contributed to pathogen clearance and immune homeostasis in infection by promoting the host protective indoleamine-2,3-dioxygenase 1/kynurenine pathway and limiting the microbial activation of the indole/aryl hydrocarbon receptor pathway. This occurred via regulation of lung and intestinal microbiota and signaling pathways. 5-HT was deficient in the sputa of patients with Cystic fibrosis, while 5-HT supplementation restored the dysregulated Trp partitioning in murine disease. These findings suggest that 5-HT, by bridging host-microbiota Trp partitioning, may have clinical effects beyond its mood regulatory function in respiratory pathologies with an inflammatory component.

Metabolism of the essential aromatic amino acid, L-tryptophan (Trp), plays an important role in mucosal homeostasis and in the pathophysiology of inflammation[1]. The endogenous metabolism of Trp mainly occurs along the kynurenine (95%), 5-hydroxytryptamine (5-HT or serotonin) (1–2%), and microbially-derived indole (4%–6%) pathways. The downstream agents of the kynurenine and indole pathways play a number of physiological roles ranging from fortifying the epithelial barrier to immune homeostasis[2]. Though often thought of as a

distinct neurotransmitter of the central nervous system (CNS), regulating mood, anxiety and behavior[3], the bioamine 5-HT is synthesized in both the CNS and peripheral tissues, mostly within the gastrointestinal tract, via the rate-limiting enzyme Trp hydroxylase (Tph) the isoform 1 (Tph1) of which is expressed in gut neural and enterochromaffin cells while the isoform Tph2 is expressed predominantly in the CNS. Mast cells (MC) are also capable of synthesizing and releasing 5-HT in mice and humans[4]. Once released, 5-HT is

[1]Department of Medicine and Surgery, University of Perugia, Perugia, Italy. [2]Istituto Zooprofilattico Sperimentale dell'Umbria e delle Marche "Togo Rosati,", Perugia, Italy. [3]CF Centre, Medical University Innsbruck, Innsbruck, Austria. [4]Division of Hygiene and Medical Microbiology, Innsbruck Medical University, Innsbruck, Austria. [5]School of Biosciences and Veterinary Medicine, University of Camerino, Camerino, Italy. [6]Department of Environmental Toxicology, University of California, Davis, CA, USA. [7]University San Raffaele and Istituto di Ricovero e Cura a Carattere Scientifico (IRCCS) San Raffaele, Rome, Italy. [8]These authors contributed equally: Emilia Nunzi, Luigina Romani. ✉e-mail: luigina.romani@unipg.it

transported into surrounding epithelial cells—or taken up by platelets in circulation—by the serotonin transporter (SERT). As 5-HT cannot readily cross the blood-brain barrier, the central and peripheral pools of 5-HT are anatomically separated and as such, act in their own distinct manners. Indeed, by acting as an autocrine and paracrine factor upon its array of receptors, 5-HT has diverse effects within the body, including platelet aggregation, bone development, and metabolic responses[3,5]. Accumulating evidence also suggests that 5-HT has an ambivalent role in the immune system, exerting both a stimulatory and inhibitory activity via its receptors[6] and/or serotonylation[7]. 5-HT recruits immune cells to the inflammatory site where it regulates both T cell-dependent and T cell-independent immune responses[8]. In the gut, 5-HT, released by enterochromaffin cells through the influence of gut microbiota[9] and short-chain fatty acids[10], can both promote and protect from inflammation[11]. Similarly in the lung, elevated 5-HT synthesis is considered an early predictor of lung injury[12], being associated with neutrophil infiltration and oxidative stress, asthma[13], pulmonary fibrosis[14], and pulmonary hypertension[15]. Of interest, platelet—and not MC—derived 5-HT is pivotal in T helper (Th)−2-mediated allergic airway inflammation[16], a finding suggesting a possible different role for MC-produced 5-HT in lung immune homeostasis.

In this regard, as reduced brain 5-HT synthesis occurs as a consequence of reduced plasma Trp availability[17], reduced 5-HT may likely occur in inflammatory conditions in which the activation of the Trp-degrading indoleamine-2,3-dioxygenase (IDO) 1 enzyme determines the flux of free Trp down the kynurenine pathway[17]. IDO1 is particularly expressed in the lung[18] where inflammatory cytokines may promote its activation along the kynurenine pathway eventually suppressing inflammation via the activation of the aryl hydrocarbon receptor (AhR)[19]. Despite the fact that Trp availability is also regulated by the tryptophan-2,3-dioxygenase (TDO)2 enzyme[20], TDO2 is mainly activated in the liver by glucocorticoids but poorly in the lung[18]. Thus, the IDO1/AhR pathway is a critical regulator of immunopathology during acute pulmonary inflammation and this may occur at the expense of 5-HT. However, because the Trp-5-HT pathway promotes IDO1 activity[21] and modulates AhR activation[22], we have hypothesized that 5-HT may have a prominent role in the interplay between the different Trp metabolic pathways.

In the present study, we have resorted to preclinical models of genetic or pharmacologically induced 5-HT deficiency or sufficiency to define the potential role of 5-HT in calibrating the IDO1/AhR axis in *A. fumigatus* pneumonia, a severe invasive infection occurring in critically ill or immunocompromised patients[23]. We found that the lung Tph1/5-HT pathway pivotally coordinates this axis in fungal pneumonia by sustaining IDO1- and restraining AhR-dependent immune activation via metabolite signaling and regulation of local and distal microbiota. Thus, 5-HT reveals a surprising protective role in lung inflammation by bridging host-microbiota interaction via Trp utilization. As a possible clinical translatability of these findings, we resorted to the Cystic fibrosis (CF) disease in which the IDO1/AhR axis and Trp metabolites have already been shown to be dysregulated[24–26]. We found a reduced serotonin level associated with a robust increase of its major metabolite, the 5-hydroxyindoleacetic acid (5-OH-IAA) in mice and humans with the p.Phe508del-CFTR mutation suggesting that targeting the Trp-5-HT pathway may have clinical effects beyond the mood and behavior to encompass the coordination of host and microbial Trp metabolic pathways in lung pathologies with an inflammatory component.

## Results

### Tph1$^{-/-}$ mice are susceptible to *A. fumigatus* infection
We have first assessed the susceptibility of Tph1$^{+/+}$ and Tph1$^{-/-}$ littermates to lung inflammation by infecting mice intranasally with *A. fumigatus* live conidia. Mice were assessed at different days post infection for parameters of infection, inflammation, and immunity. We

found a higher fungal burden in the lung of Tph1$^{-/-}$ than Tph1$^{+/+}$ littermates with signs of dissemination into the brain and the presence of fungal hyphae (Fig. 1a) associated with recruitment of monocytes and increased expression of monocyte-recruiting chemokines at the expenses of neutrophil recruitment and *Cxcl1* expression (Fig. 1b–d and Table S1). Phagocytosis and not the conidiocidal or oxidative activity was impaired in recruited monocytes (Fig. 1e), consistent with the ability of 5-HT to stimulate phagocytosis in macrophages[8]. Together with the defective neutrophil recruitment, whose phagocytosis was also defective (Fig. 1e), this result may explain the unrestricted fungal growth in the relative absence of 5-HT. The histological analysis revealed a progressive increased inflammatory pathology (Fig. 1f, g) and damage (Fig. 1h) in the lung of Tph1$^{-/-}$ compared to Tph1$^{+/+}$ mice in which the early inflammatory reaction is progressively subsiding (Fig. 1f–h). In addition to the defective innate response, a decreased Th1, Th2 and regulatory T (Treg) cell expansion and an increased Th17 cell expansion in Tph1$^{-/-}$ mice was observed, as revealed by the Th cytokine levels in lung homogenates and the expression of Th transcription factors in the draining lymph nodes (Fig. 1i). Of interest, IL-9, known to mediate immune tolerance via Tph1-expressing MC-Treg crosstalk[27], was also defective in Tph1$^{-/-}$ mice along with the Th-9 transcription factor PU.1 (Fig. 1i). Together, these findings indicate that both the innate and adaptive immune responses are defective in Tph1$^{-/-}$ mice with fungal pneumonia and are fully consistent with the ability of 5-HT to recruit neutrophils to sites of acute inflammation[28] and with the defective T cell priming[8] and tolerance[29] observed in Tph1$^{-/-}$ mice.

To causally link 5-HT deficiency to the findings observed in Tph1$^{-/-}$ mice, we administered the Tph1 inhibitor p-chlorophenylalanine (PCPA), known to deplete 5-HT[30], and the 5-HT precursor (5-HTP), to Tph1$^{+/+}$ or Tph1$^{-/-}$ infected mice, respectively (Fig. 2a). We found that 5-HT depletion in Tph1$^{+/+}$ mice increased the fungal growth (Fig. 2c), decreased neutrophil recruitment (Fig. 2d), increased inflammatory pathology (Fig. 2e) and inflammatory cytokine production (Fig. 2f), thus recapitulating the findings observed in Tph1$^{-/-}$ mice. In contrast, 5-HT repletion increased survival (Fig. 2b), restricted fungal growth, and ameliorated lung pathology and inflammatory cytokine production, thus improving resistance to infection in Tph1$^{-/-}$ mice (Fig. 2b–f). Together, these findings show that genetic or pharmacologic inhibition of 5-HT significantly increases susceptibility to fungal pneumonia. Consistent with the reduced allergic airway inflammation as a result of defective ILC2[31] and T cell priming described in Tph1$^{-/-}$ mice, we found a decreased susceptibility of Tph1$^{-/-}$ mice to fungal allergy (Fig. S1), a finding pointing to a unique, as yet unappreciated, beneficial role 5-HT may have in acute pneumonia.

### Mast cells produce 5-HT in *A. fumigatus* infection
To test whether 5-HT is produced in the lung during the infection, the expression of genes encoding the enzymes involved in the 5-HT synthesis and degradation (Fig. 3a) as well as the 5-HT levels were assessed in total lung cells and lung homogenates, respectively, from Tph1$^{+/+}$ and Tph1$^{-/-}$ littermates at different time points after the infection. The gene expression of both the first rate-limiting *Tph1* and *Ddc* enzymes of 5-HT synthesis, known to correlate with 5-HT production in the lung[12], were increased early in infection in Tph1$^{+/+}$ mice and followed later on by the increased expression of the genes *Aanat* and *Maoa*, encoding for 5-HT degradation to N-acetyl-hydroxytryptamine (NAS) and 5-OH-IAA[21,32], respectively, but not of the *Asmt* gene encoding for melatonin production (Fig. 3b). Also increased was the expression of the *Sert* gene encoding for the serotonin transporter, but not that of *Tph2* encoding gene (Fig. 3b). Accordingly, significant increase levels of the 5-HTP and 5-HT were observed in the lung of Tph1$^{+/+}$ mice early in infection to decline thereafter (Fig. 3c). Consistent with the diurnal regulation of *Tph1*[33], the production of 5-HT was even higher at night (74.14 (night) *vs* 7.06 pg/µg protein (day); $p = 0.0143$). Although not increased in infection, low levels of 5-HTP and 5-HT could

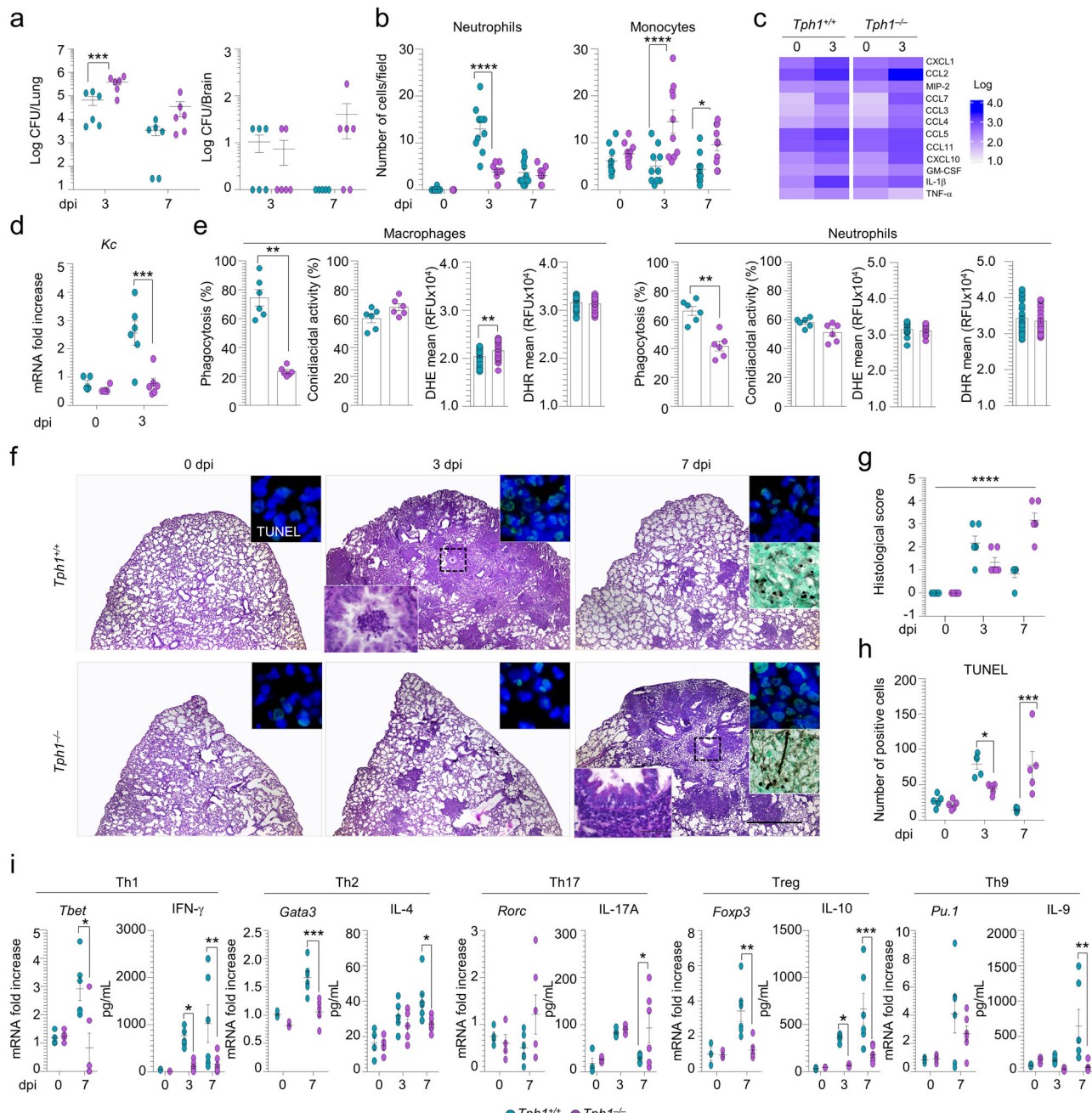

**Fig. 1 | *Tph1$^{-/-}$* mice are susceptible to *A. fumigatus* infection.** *Tph1$^{+/+}$* and *Tph1$^{-/-}$* littermates were infected intranasally with *A. fumigatus*. Mice were sacrificed at 3 or 7 days post infection (dpi) and monitored for **a** fungal growth (*n* = 6 mice/group), **b** number of neutrophils and monocytes (*n* = 10 fields over 6 mice/group) in the bronco-alveolar lavage, **c, d** lung inflammatory cytokine and chemokine expression (**c**, *n* = 2; **d**, *n* = 4–6 mice/group), **e** in vitro effector activity (*n* = 6 mice/group) and reactive oxygen species production of ex-vivo lung macrophages (DHE and DHR, *n* = 39 readings over six mice/group) and neutrophils (DHE, *n* = 13 and DHR, *n* = 28 readings over 6 mice/group) from uninfected mice, **f** lung histology (periodic acid-Schiff staining and, in the insets, TUNEL and Gomori methenamine-silver stainings), **g** histological score (*n* = 6 mice/group) and **h** number of TUNEL-positive cells (*n* = 5 mice/group). **i** T helper-related gene expression in thoracic lymph nodes (RT-PCR) and cytokine production (ELISA) (*n* = 3–7 mice/group). Hoechst was used for nuclear counterstain in blue. Photographs were taken with a high-resolution

microscope (Olympus BX51), ×4 and ×100 magnification (scale bars 1000 μm and 20 μm, respectively). Data are presented as mean ± SEM and representative of three experiments. *$p < 0.05$, **$p < 0.01$, ***$p < 0.001$, and ****$p < 0.0001$, *Tph1$^{-/-}$ vs Tph1$^{+/+}$* mice. **a, b, d** Two-way ANOVA, Sidak post test. Adjusted *P* values for **a**: 0.0003; **b**, left panel: <0.0001; **b**, right panel: <0.0001 (3 dpi), 0.0313 (7 dpi); **d**: 0.0003. **e** Mann–Whitney test, two-tailed. *P* value for phagocytosis: 0.0022 (Macrophages), 0.0043 (Neutrophils); Unpaired *t* test, two-tailed. *P* value for DHE: 0.0032 (Macrophages), **g** Kruskal–Wallis test. *P* value: <0.0001. **h** Two-way ANOVA, Sidak post test. Adjusted *P* values: 0.0291 (3 dpi), 0.0001 (7 dpi); **i** Two-way ANOVA, Sidak post-test. Adjusted *P* values: 0.024 (*Tbet*), 0.0343 (IFN-γ, 3 dpi), 0.0037 (IFN-γ, 7 dpi), 0.0001 (*Gata3*), 0.0458 (IL-4), 0.0212 (IL-17A), 0.0027 (*Foxp3*), 0.0366 (IL-10, 3 dpi), 0.0005 (IL-10, 7 dpi), 0.0023 (IL-9). Source Data are provided as a Source Data file.

be detected in *Tph1$^{-/-}$* mice, likely occurring through a phenylalanine hydroxylase-dependent pathway as described[34]. Of note, 5-HT levels increased in the lung but not in the serum, suggesting that 5-HT could be locally produced. This appeared to be the case as the early increase

of platelets and 5-HT immunostaining in the lung of *Tph1$^{+/+}$* (Fig. S2), as opposed to *Tph1$^{-/-}$* littermates (Fig. 3d and Table S2), was greatly reduced but still present upon lung perfusion (Fig. 3d). Immunostaining revealed that MC and not pulmonary neuroepithelial bodies,

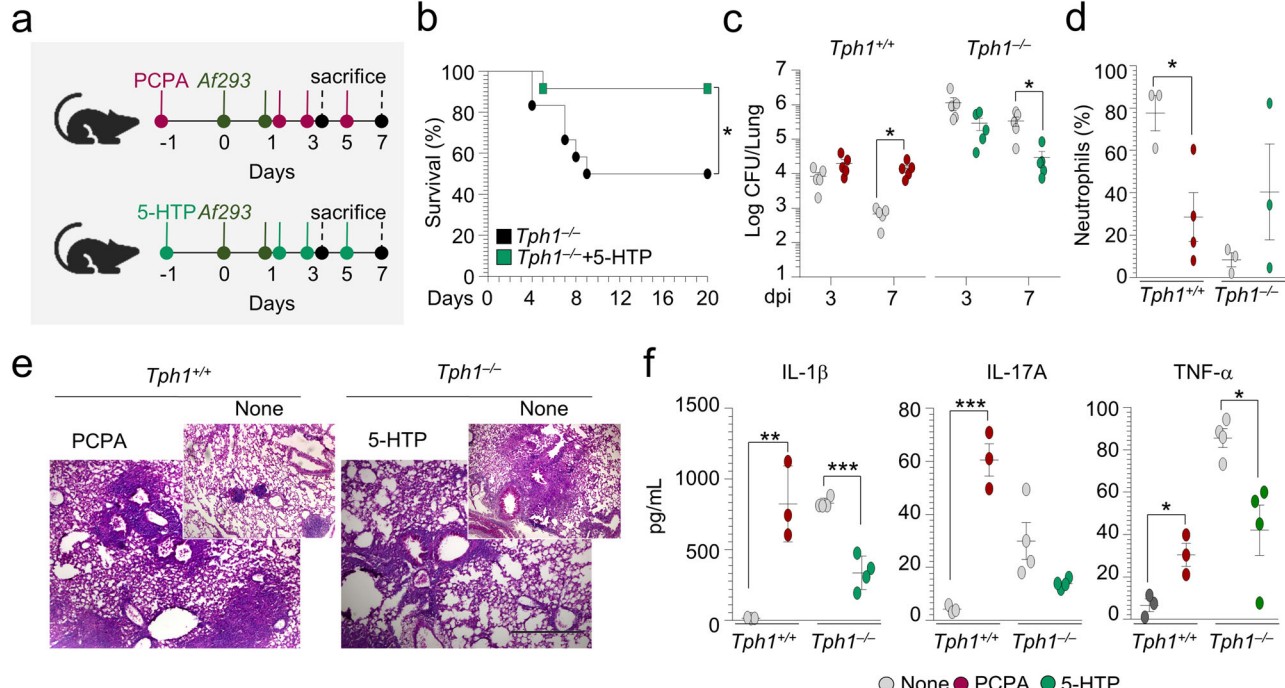

**Fig. 2 | Effects of PCPA and 5-HTP administration in infection. a** *Tph1*⁺/⁺ and *Tph1*⁻/⁻ littermates were infected intranasally with *A. fumigatus* and treated intra-peritoneally with 5.5 mg/mouse of parachlorophenylalanine (PCPA) or 5-Hydroxytryptophan (5-HTP), respectively, 1 day before the infection and every other day post infection (dpi). Mice were monitored for **b** survival (%) (*n* = 12 mice/group), **c** fungal burden (*n* = 5 mice/group), and **d** neutrophils (%) in the bronco-alveolar lavage (*n* = 3–4 mice/group), **e** lung histology (periodic acid-Schiff stain-ing) and **f** cytokine production in lung homogenates (ELISA) at 7 dpi (*n* = 3 in *Tph1*⁺/⁺

and *n* = 4 in *Tph1*⁻/⁻ mice/group). Photographs were taken with a high-resolution microscope (Olympus BX51), ×10 magnification (scale bar 500 μm). Data are expressed as mean ± SEM and are representative of two experiments. *$p < 0.05$, **$p < 0.01$, and ***$p < 0.001$. **b** Log-rank (Mantel-Cox) test. *P* value 0.0319. **c**–**f** Unpaired *t* test, two-tailed. *P* values in **c**: 0.0300 (*Tph1*⁺/⁺), 0.017 (*Tph1*⁻/⁻); in **d**: 0.0236; in **f**: for IL-1β, 0.0065 (*Tph1*⁺/⁺), 0.0002 (*Tph1*⁻/⁻); for IL-17A, 0.0008; for TNF-α, 0.0185 (*Tph1*⁺/⁺), 0.0141 (*Tph1*⁻/⁻). Source Data are provided as a Source Data file.

known to produce 5-HT[35], released 5-HT (Fig. 3d). Consistently, 5-HT-producing MC were not observed in the lung of either MC-deficient or AhR-deficient infected mice, consistent with the notion that AhR modulates MC responses[36] (Fig. S2). Thus, in addition to platelets recruited in infection, MC produces 5-HT in the lung via AhR.

## 5-HT coordinates the IDO1/AhR axis in lung pneumonia

The bulk of investigations regarding 5-HT suggest that 5-HT mediates pro-inflammatory responses. Less is known about the ability of 5-HT to regulate Trp utilization in inflammation. To address this question, we first assessed the activation of the IDO1- and AhR-dependent pathways in *Tph1*⁺/⁺ and *Tph1*⁻/⁻ mice with fungal pneumonia and the influence of PCPA or 5-HTP administration on it. In line with previous findings[24], IDO1 protein expression increased in infection in *Tph1*⁺/⁺ lung as revealed by immunofluorescence staining (Fig. 4a and Table S2), immunoblotting (Fig. 4b) and expression of genes encoding the IDO1-downstream enzymes *Haao* and *Kmo* (Fig. 4c). We also looked for the expression of the *Tdo2* gene, for its major role in determining Trp availability[20], and did not find its activation in both the lung and liver during the infection (Fig. 4d). Neither *Ido1* nor *Tdo2* activation was observed in *Tph1*⁻/⁻ mice (Fig. 4a–d). In contrast, a progressively increasing AhR activation was observed in these mice, as revealed by immunofluorescence staining (Fig. 4e and Table S2), immunoblotting (Fig. 4f), expression of AhR-dependent genes (Fig. 4g) and AhR activity in the lung (Fig. 4h). As already shown[25], AhR activation occurred in C57BL/6 mice early in infection to subside thereafter (Fig. 4e–h). PCPA administration decreased the IDO1 pathway and promoted AhR acti-vation in *Tph1*⁺/⁺ whereas 5-HTP administration restored the IDO1 pathway and restrained AhR activation in *Tph1*⁻/⁻ mice (Fig. 4i), thus revealing a pivotal role of 5-HT in Trp partitioning during lung infec-tion. Importantly, PCPA administration failed to exacerbate infection

and immunopathology in condition of AhR blockade, whereas 5-HTP administration failed to improve resistance to infection and pathology in condition of IDO1 blockade (Fig. S3). Together, these findings sug-gest that an imbalanced Trp utilization occurred in *Tph1*⁻/⁻ mice whereby a defective IDO1 pathway is associated with an unrestrained AhR activation and that 5-HT acts upon this balance. Because Tph1 shares a similar Km for Trp (~20 μM)[37] as IDO1, it can potentially exhaust Trp to regulate immune tolerance. This points to alternative mechanisms of defective IDO1 expression in *Tph1*⁻/⁻ mice.

## Microbiota and metabolites are altered in the lung of Tph1⁻/⁻ mice

It has been suggested that the metabolite NAS, in the serotonin path-way, directly binds IDO1 and acts as a positive allosteric modulator of the IDO1 enzyme, thus orchestrating the interplay between the two Trp metabolic pathways[21]. However, the microbiota also contributes to the balance among the different Trp metabolic pathways by regulating the bioavailability of Trp[38] and the production of bioactive metabolites regulating tissue immune homeostasis[39] via a number of circuits including the bidirectional microbiome-AhR axis[39,40]. We, therefore, resorted to 16S rRNA sequencing after rarefaction strategies (Fig. S4) to compare microbial composition between *Tph1*⁻/⁻ and *Tph1*⁺/⁺ litter-mates in order to highlight both qualitative and quantitative differ-ences in lung community proportions. The most represented phyla in the lung of uninfected *Tph1*⁺/⁺ littermates were *Firmicutes* (46.3%), *Actinobacteriota* (34.3%), *Proteobacteria* (15.7%) and *Bacteroidota* (3.1%), a phylum distribution different from that observed in *Tph1*⁻/⁻ mice in which *Proteobacteria* (58.8%), mainly the Gammaproteo-bacteria class and the *Helicobacter* genus, were expanded at the expenses of *Firmicutes* (7.9%) (Figs. 5a and 6). The analysis of α-diversity did not reveal significant differences in richness and evenness

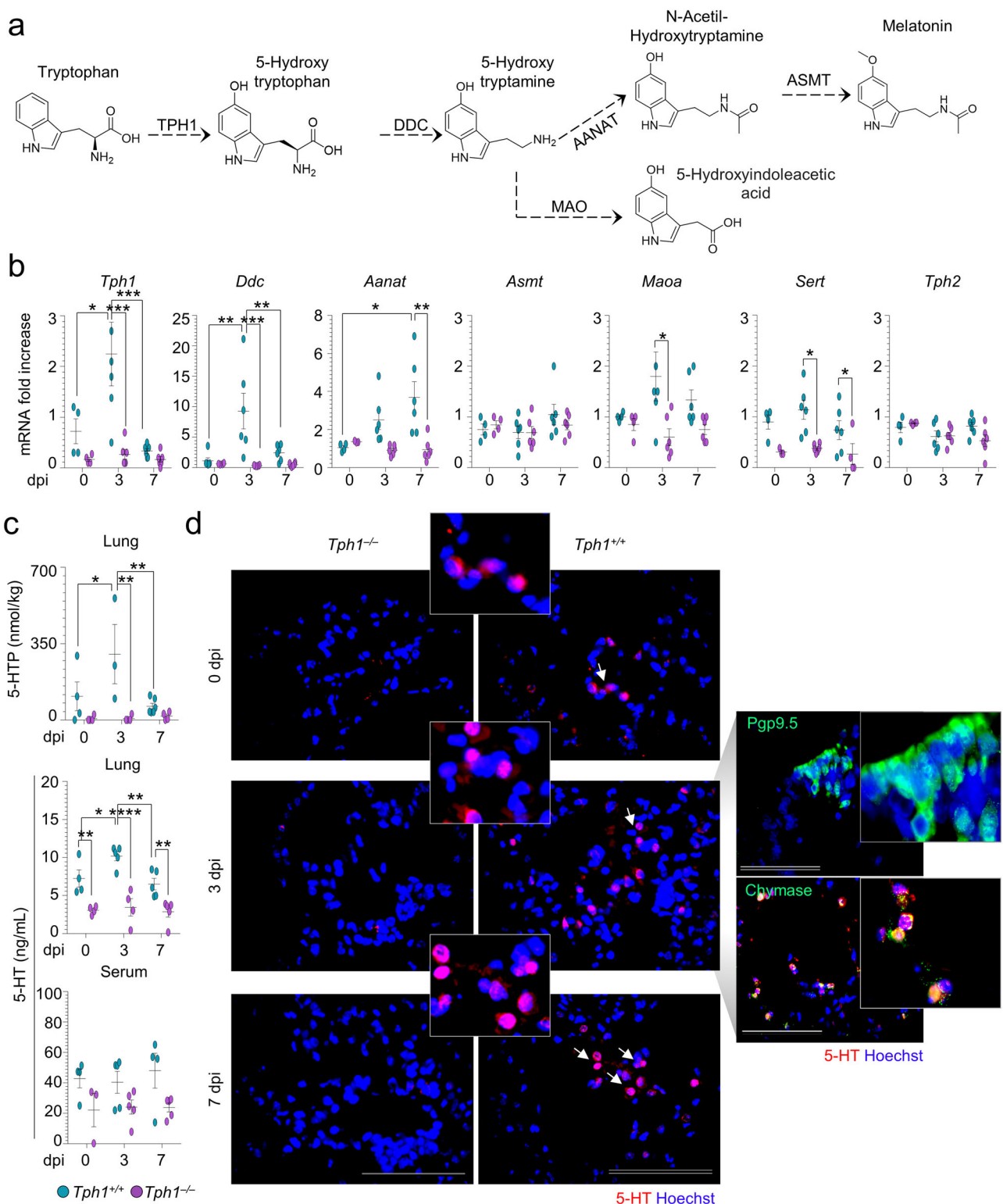

between *Tph1⁻/⁻* and *Tph1⁺/⁺* littermates, as measured by observed operational taxonomic units, Chao1 and Shannon indexes ($p > 0.05$ for all indexes). On the contrary, for each dpi point, the two types of mice differed in compositional structure by β-diversity and were separated into distinct clusters in infection, as shown by visualization of the Jaccard and Bray-Curtis indexes, the principal coordinate ordination (PCoA), both at the sequence variant level (Fig. 5b and Table S3), the linear discriminant analysis effect size (LEfSe) at the genus level and the ANOVA-like Differential Gene Expression Analysis (Q2-ALDEx2) at the sequence variant level (Fig. 6). Specifically, *Streptococcaceae*

(*Lactococcus* genus) and *Verrucomicrobia* (*Akkermansia* genus) expanded early, at 3 dpi, along with *Gammaproteobacteria (E. coli/ Shigella* genus*)* in *Tph1⁺/⁺* littermates followed by the expansion of *Bacteroidota (Marinifilaceae, Rikenellaceae, Odoribacter and Alistipes )* at the expenses of *Gammaproteobacteria* at 7 dpi, in concomitance with infection resolution, as already shown[41]. In contrast, *Propionibacteriaceae* (*Cutibacterium* genus*), Alcaligenaceae,* and Enterobacteriaceae expanded early followed by the expansion of *Sphingomonadaceae* (*Sphingomonas* genus) late in infection in *Tph1⁻/⁻* littermates.

**Fig. 3 | Mast cells produce 5-HT in *A. fumigatus* infection. a** The tryptophan hydroxylase (TPH)1 pathway (DDC, dopa decarboxylase; AANAT, arylalkylamine N-acetyltransferase; ASMT, N-acetylserotonin O-methyltransferase; MAO, mono-amine oxidase; ALDH, aldehyde dehydrogenase). *Tph1*$^{+/+}$ and *Tph1*$^{-/-}$ littermates were infected intranasally with *A. fumigatus* and assessed at 3 or 7 days post infection (dpi) for **b** lung gene expression (RT-PCR) (*n* = 4–6 mice/group), **c** levels of 5-HTP and 5-HT in perfused lungs and serum (*n* = 3–6 mice/group) and **d** immunofluorescence staining of 5-HT, pulmonary neuroepithelial bodies (Pgp9.5) and mast cells (chymase) in perfused lungs. White arrows indicate 5-HT$^+$ cells. Hoechst was used for nuclear counterstain in blue. Photographs were taken with a high-resolution microscope (Olympus BX51), ×100 magnification (scale bars 20 μm). Data are expressed as mean ± SEM and representative from three

experiments. *$p < 0.05$, **$p < 0.01$, ***$p < 0.001$, and ****$p < 0.0001$. Two-way ANOVA, Tukey post hoc test. Adjusted *P* values for **b**: for *Tph1*, 0.0273 (*Tph1*$^{+/+}$, 0 vs 3 dpi), 0.0010 (*Tph1*$^{+/+}$, 3 vs 7 dpi), 0.0007 (3 dpi, *Tph1*$^{+/+}$ vs *Tph1*$^{-/-}$); for *Ddc*, 0.0013 (*Tph1*$^{+/+}$, 0 vs 3 dpi), 0.0084 (*Tph1*$^{+/+}$, 3 vs 7 dpi), 0.0004 (3 dpi, *Tph1*$^{+/+}$ vs *Tph1*$^{-/-}$); for *Aanat*, 0.0121 (*Tph1*$^{+/+}$, 0 vs 7 dpi), 0.0027 (7 dpi, *Tph1*$^{+/+}$ vs *Tph1*$^{-/-}$); for *Maoa*, 0.0217 (3 dpi, *Tph1*$^{+/+}$ vs *Tph1*$^{-/-}$); for *Sert*, 0.0131 (3 dpi, *Tph1*$^{+/+}$ vs *Tph1*$^{-/-}$), 0.0343 (7 dpi, *Tph1*$^{+/+}$ vs *Tph1*$^{-/-}$); for **c**: for 5-HTP, 0.0019 (3 dpi, *Tph1*$^{+/+}$ vs *Tph1*$^{-/-}$), 0.0055 (*Tph1*$^{+/+}$, 3 vs 7 dpi), 0.038 (*Tph1*$^{+/+}$, 0 vs 3 dpi); for lung 5-HT, 0.0069 (0 dpi, *Tph1*$^{+/+}$ vs *Tph1*$^{-/-}$), < 0.0001 (3 dpi, *Tph1*$^{+/+}$ vs *Tph1*$^{-/-}$), 0.0086 (7 dpi, *Tph1*$^{+/+}$ vs *Tph1*$^{-/-}$), 0.0071 (*Tph1*$^{+/+}$, 3 vs 7 dpi), 0.0456 (*Tph1*$^{+/+}$, 0 vs 3 dpi). Source Data are provided as a Source Data file.

We then resorted to Phylogenetic Investigation of Communities by Reconstruction of Unobserved States 2 (PICRUSt2) and the KEGG database to determine the presence and relative abundance of functional modules inferred from 16S data. As shown in Fig. 7 and S5 and Table S4, a number of metabolic pathways were differently expressed in the two types of mice at the basal level and in infection. The LPS biosynthesis, sulfur reduction, pectin degradation, ketone body biosynthesis, and putrescin biosynthesis modules were all increased as opposed to the decreased glycolysis, gluconeogenesis, Krebs cycle, and fatty acid biosynthesis in *Tph1*$^{-/-}$ as compared to *Tph1*$^{+/+}$ littermates (Fig. S5). In particular, given the crucial role of not only host but also microbial Trp metabolic enzymes in infection[42], we were much intrigued by the findings that the biosynthetic pathways of aromatic amino acids, including the Shikimate, anthranylate and Trp synthase enzymatic pathways, all increased in either type of mice but particularly in *Tph1*$^{-/-}$ littermates, indicating active Trp biosynthesis in infection (Fig. 7a, b). Along with Trp biosynthesis, microbial Trp catabolism was also increased in *Tph1*$^{-/-}$ as compared to *Tph1*$^{+/+}$ littermates, as revealed by the progressive upregulation of both the indole (tryptophanase, monoamine oxidase and aldehyde dehydrogenase) and the kynurenine (kynurenine monooxygenase, kynunerinase, hydroxyanthranilate dioxygenase and anthranilate monooxygenase) pathways (Fig. 7b and Table S4). Accordingly, the level of Trp decreased (Fig. 7c) while those of the indole pathway (including the Indole-3-propionic acid, Indole-3-lactic acid, Indole-3-carboxaldehyde, Indole-3-carboxylic acid, and Indoxyl sulfate) (Fig. 7d) increased in infection in *Tph1*$^{-/-}$ mice, whereas the levels of metabolites along the kynurenine pathway (Kynurenic acid and Xanthurenic acid), although increased in infection, were not significantly different from those detected in *Tph1*$^{+/+}$ littermates, except for kynurenines (Fig. 7e). This finding suggests the contribution of kynurenine-producing bacteria in these mice (see discussion). Consistent with the defective 5-HT, neither NAS nor 5-OH-IAA was detected in *Tph1*$^{-/-}$ mice as opposed to *Tph1*$^{+/+}$ littermates (Fig. 7f). Considering that, in addition to NAS[21], 5-OH-IAA was also able to activate the AhR/IDO1 pathway in vitro (Fig. S6), together, these results indicate that the relative presence or absence of Trp metabolites with distinct signaling functions may explain the sustained AhR and the defective IDO1 activation in condition of 5-HT deficiency.

**The fecal microbiota is altered in Tph1$^{-/-}$ mice**

While the above results clearly implicate a role for the lung microbiota in the immunometabolic profile during lung pneumonia, they do not address the role, if any, of the intestinal microbiota. Considering the bidirectional crosstalk between the gut and the lungs via the microbiota and metabolites[43], and the multiple activity of 5-HT in the gut, including the regulation of bacterial growth[44], we have evaluated the fecal microbial composition and function in *Tph1*$^{-/-}$ and *Tph1*$^{+/+}$ littermates in infection. *Firmicutes, Bacteriodota, Proteobacteria,* and *Actinobacteriota* were the most represented phyla in both types of uninfected mice, with the exception of the *Verrucomicrobia* and the *Bacteroidota* taxa that were more expanded in *Tph1*$^{+/+}$ and *Tph1*$^{-/-}$

littermates, respectively (Fig. 5c, d). As in the lung, there was no significant difference in the α-diversity between the two types of mice as calculated by operational taxonomic units, Chao1 and Shannon diversity indexes (*p* > 0.05 for all indexes). However, the two strains of mice differed for β-diversity, being separated into distinct clusters before and during the infection, as shown by visualization of the Jaccard and Bray-Curtis indexes and the principal coordinate ordination (PCoA), both at the sequence variant level (Fig. 5d and Table S3), the linear discriminant analysis effect size LEfSe at the genus level and the Q2-ALDEx2 at the sequence variant level (Fig. 8). Specifically, the first component of both Jaccard and Bray-Curtis indexes showed significant differences between mice at each dpi (*p* adj ≤ 0.05), thus revealing both qualitative and quantitative differences of microbial composition between mice independently from dpi. LEfSe and Q2-ALDEx2 confirmed the relative expansion of Verrucomicrobia (Akkermansia genus), Desulphobacteriota (Desulfovibrio genus) and *Firmicutes* (*Lactobacillus, Lactococcus, Ruminococcus, Turicibater, Blautia* and *Clostridia sensu_strictu* genera, among others) in uninfected *Tph1*$^{+/+}$ littermates as opposed to the high abundance of *Bacteroidota* (*Bacteroides, Alistipes, Rikenella genera* among others), *Proteobacteria* (*Parasutterella* genus), *Actinobacteriodota* (*Parvibacter* genus) and Campilobacterota (Helicobacter genus) in *Tph1*$^{-/-}$ littermates (Fig. 8). As in the lung, *Proteobacteria* (*Escherichia-Shigella* genus) expanded early in infection in *Tph1*$^{+/+}$ mice along with *Firmicutes* (*Enterococcus, Lactococcus* and *Clostridium sensu_strictu* genera, among others), the *Desulfovibrio* genus and *Bacteriodota* (*Parabacteroides* genus) and subsided late in concomitance with the expansion of *Firmicutes* (*Lachnospiranaceae_NK4A136_group, Turicibacter, Duboisella* and *Lactococcus* genera, among others) and Bacteriodota (Prevotellaceae_NK3B31_group and Rikenella genera). In contrast, *Bacteriodota* (*Bacteroides* and *Prevotella_UCG_OO2* genera) and *Burkorderiales* (*Parasutterella* genus) expanded early in *Tph1*$^{-/-}$ littermates followed by the unrestricted expansion of *Burkhorderiales* (*Parasutterella* genus), along with *Actinobacteriota* (*Enterorabdhus* and *Parvibacter* genera) and *Bacteriodota* (*Bacteroides* and *Alistipes* genera) (Fig. 8). In terms of functional modules, the biosynthetic pathways of the aromatic amino acids phenylalanine and tyrosine, but not Trp, was in general higher in *Tph1*$^{-/-}$ than *Tph1*$^{+/+}$ littermates (Fig. 7g). For Trp, indeed, the degrading more than the biosynthetic pathway was apparently activated in either type of mice, as revealed by the relative expression of the aldehyde dehydrogenase, kynureninase and, particularly in *Tph1*$^{-/-}$ mice, the tryptophanase enzymes (Fig. 7g and Table S4). As in the lung, the LPS biosynthesis, pectin degradation, mannose, and lactose transport system, ketone body biosynthesis, and putrescin biosynthesis modules were all increased in *Tph1*$^{-/-}$ as compared to *Tph1*$^{+/+}$ littermates (Fig. S7). Despite no differences in the glycolysis and Krebs cycle enzymatic pathways, the expression of the fumarate reductase, succinate dehydrogenases, and glyoxylate cycle enzymes were increased in *Tph1*$^{-/-}$ mice (Fig. S7), suggesting neoglucogenesis optimization. Network analysis of lung and gut microbiota confirmed the association of genera and sequence variants identified by LEfSe and Q2- ALDEx2 with the activation of the indole and

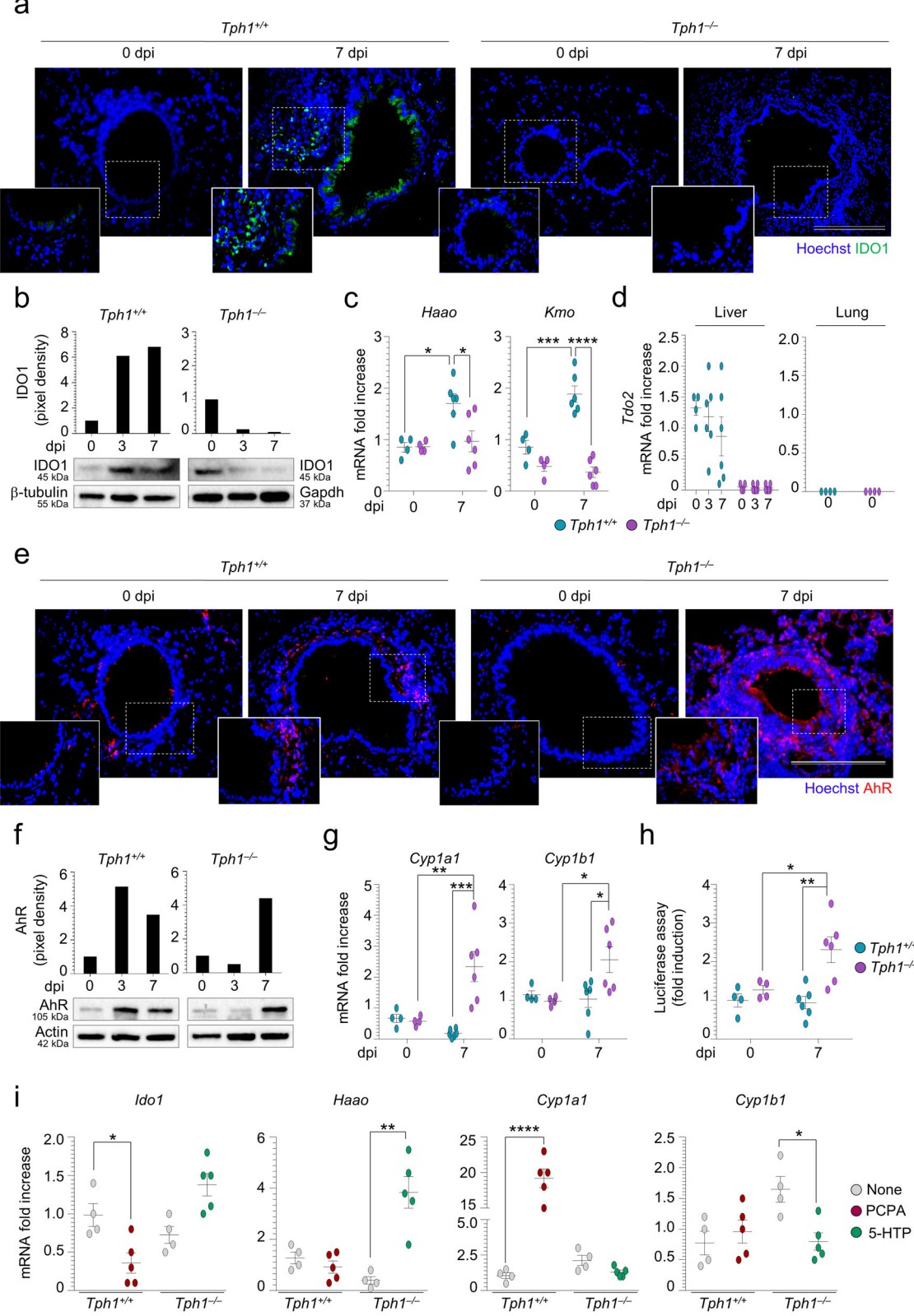

kynurenine enzymes of Trp degradation pathways in infection, mostly and variably contributed by *Firmicutes* and *Bacteriodota*, and partly *Actinobacteria*, in each site, with the interesting finding of Proteobacteria (*Sphingomonas* genus) contributing to the kynurenine degrading pathway in the lung (Fig. 9a). Of interest, enzymes of the 5-HT biosynthesis pathway were not observed (Fig. 9a). Overall, these results suggest that, despite similarities between the lung and fecal microbiota composition in *Tph1⁻/⁻* mice, differences appeared to be

present at the level of carbohydrate utilization but more Trp utilization, as revealed by the higher expression of microbial Trp-degrading enzymes in *Tph1⁻/⁻* littermates (Fig. 9b).

## The local and distal microbiota contribute to the IDO1/AhR axis in the lung

To directly assess the contribution of the local *vs* the distal microbiota to the Trp metabolism in infection, we did criss-cross experiments in

**Fig. 4 | 5-HT coordinates the IDO1/AhR axis in lung pneumonia.** $Tph1^{+/+}$ and $Tph1^{-/-}$ littermates were infected intranasally with *A. fumigatus* and lungs were assessed at 3 or 7 days post infection (dpi) for **a** IDO1 expression by immuno-fluorescence staining and **b** western blotting, **c** *Haao* and *Kmo* expression, **d** *Tdo2* expression (also in the liver), **e** AhR expression by immunofluorescence staining and **f** western blotting, **g** *Cyp1a1* and *Cyp1b1* expression and **h** AhR activity (luci-ferase assay on H1L6.1 cells exposed to lung homogenates from infected mice). **c**, **d**, **g**, **h** ($n = 4$ mice/group in 0 dpi and $n = 6$ mice/group in 3 and 7 dpi), **i** Lung gene expression in mice treated with parachlorophenylalanine (PCPA) or 5-Hydroxytryptophan (5-HTP) as in legend to Fig. 2 ($n = 4$ mice/group in None and $n = 5$ mice/group in treated mice). Hoechst was used for nuclear counterstain in blue. Photographs were taken with a high-resolution microscope (Olympus BX51),

×40 and ×100 magnification (scale bars 100 and 20 μm, respectively). Data are expressed as mean ± SEM and representative from three experiments. *$p < 0.05$, **$p < 0.01$, ***$p < 0.001$, and ****$p < 0.0001$, $Tph1^{-/-}$ *vs* $Tph1^{+/+}$ mice, infected vs uninfected (none) mice, treated vs untreated mice. One-way or two-way ANOVA, Turkey post hoc test. Adjusted *P* values for **c**: for *Haao*, 0.0196 ($Tph1^{+/+}$, 0 vs 7 dpi), 0.0249 (7 dpi, $Tph1^{+/+}$ vs $Tph1^{-/-}$); for *Kmo*, 0.0003 ($Tph1^{+/+}$, 0 vs 7 dpi), <0.0001 (7 dpi, $Tph1^{+/+}$ vs $Tph1^{-/-}$); for **g**: for *Cyp1a1*, 0.0055 ($Tph1^{-/-}$, 0 vs 7 dpi), 0.0003 (7 dpi, $Tph1^{+/+}$ vs $Tph1^{-/-}$); for *Cyp1b1*, 0.0384 ($Tph1^{-/-}$, 0 vs 7 dpi), 0.0265 (7 dpi, $Tph1^{+/+}$ vs $Tph1^{-/-}$); for **h**: 0.0381 ($Tph1^{-/-}$, 0 vs 7 dpi), 0.0021 (7 dpi, $Tph1^{+/+}$ vs $Tph1^{-/-}$). **i** Unpaired *t* test, two-tailed, *P* values in treated vs None mice: for *Ido1*, 0.0165 ($Tph1^{+/+}$), 0.0114 ($Tph1^{-/-}$); for *Haao*, 0.0019 ($Tph1^{-/-}$); for *Cyp1a1*, <0.0001 ($Tph1^{+/+}$), for *Cyp1b1*, 0.0103 ($Tph1^{-/-}$). Source Data are provided as a Source Data file.

which the lung or fecal microbiota from $Tph1^{+/+}$ was transplanted into $Tph1^{-/-}$ littermates and vice versa. We found that lung microbiota transplantation significantly altered immune and metabolic para-meters in infection. In particular, transplantation of $Tph1^{+/+}$ lung microbiota restored resistance to infection and decreased inflamma-tion in $Tph1^{-/-}$ mice while the opposite was true in $Tph1^{+/+}$ littermates transplanted with $Tph1^{-/-}$ lung microbiota (Fig. 9c–f). In terms of Trp metabolism, transplantation of $Tph1^{+/+}$ lung and fecal microbiota pro-moted the expression of IDO1-dependent genes in $Tph1^{-/-}$ lung (Fig. 9g), whereas, consistent with the high content of AhR ligands, transplantation of $Tph1^{+/+}$ lung or fecal microbiota further increased the expression of AhR-dependent genes in $Tph1^{-/-}$ littermates, (Fig. 9g). Of interest, no effects were observed on the IDO1/AhR expression in transplanted $Tph1^{+/+}$ littermates, a finding indicating the tightly-regulated IDO1/AhR expression in these mice. Together, these results indicate that the local microbiota actively participates in shaping the immune and metabolic activity of the lung in fungal pneumonia in the condition of 5-HT deficiency. The results also showed that the intest-inal microbiota is not only sensitive to the infection but likely con-tributes to the lung immunometabolic activity during the infection.

## The lung Tph1/IDO1/AhR axis is altered in cystic fibrosis

To grasp the possible clinical relevance of the above findings, we have resorted to the CF disease because the IDO1/AhR axis has already been shown to be dysregulated[24,25], the down-regulation of the CFTR abol-ishes the hypoxia-induced 5-HT release[45] and 5-HT is among the top significant metabolites discriminating the metabolic profiles among CF mutations and lung function[46]. To this purpose, we have evaluated the expression of genes encoding the enzymes involved in the 5-HT synthesis and degradation in the lung of $Cftr^{F508del}$ mice at different time points after the infection. We also assessed the 5-HT levels in the lung of CF mice and sputa from patients with CF. We found an increased expression of *Aanat* and *Maoa* genes encoding for 5-HT-degrading enzymes more than the rate-limiting *Ddc* enzyme (Fig. 10a). Accord-ingly, significant decreased 5-HT levels were observed in the lung of $Cftr^{F508del}$ mice, as revealed by specific ELISA (Fig. 10b) and immunos-taining (Fig. 10c), associated with a significant increase of 5-OH-IAA (Fig. 10b), indicating active 5-HT catabolism. Significant decreased levels were also observed in the sputa of patients with the $\Delta F508$ homozygous mutation (Fig. 10d). As 5-HT repletion by exogenous administration of 5-HTP improved resistance to infection and inflam-mation in $Cftr^{F508del}$ mice (Fig. 10e–g) and promoted IDO1/AhR activity balance (Fig. 10h), these results suggest that the benefit of 5-HT restoration in CF may go beyond mood control to include an action on Trp partitioning and immune homeostasis in the lung.

## Discussion

The association of dysregulated Trp metabolism with a variety of chronic diseases suggests its exploitation for therapy[47]. This demands for a thorough dissection of the intricacies and interplay among the different metabolic pathways in the different clinical settings for the successful co-option of signaling metabolites for therapy. In the

present work, we demonstrated that the Tph1/5-HT pathway, by criti-cally regulating Trp partitioning in the lung, profoundly affects the local innate and adaptive immune response in infection, thus con-firming the critical role of Trp in regulating lung immunity and inflammation[19,48]. Both the pathogen growth and the associated inflammatory pathology were poorly controlled in the relative absence of 5-HT, a finding consistent with the well-known activity of 5-HT on innate and adaptive immune cells[6]. Here, however, it is also worth mentioning that platelets release 5-HT on contact with fungi[48] and that both 5-HT[49] and serotonin reuptake inhibitors (SSRIs)[50] possess intrinsic antifungal activity. The findings that $Tph1^{-/-}$ mice are also susceptible to gastrointestinal[51] and vaginal (Fig. S8) candidiasis, while further pointing to the sensitivity of fungi to local changes of 5-HT, lead to a conceptualization that among the multitasking activity of 5-HT is an integrated physiological and behavioral response to inflammatory events and pathogens at the host-pathogen interface. In the lung, this may occur upon accumulation via platelets or by local production by host cells and not, apparently, by microbes. Indeed, MC were found to accumulate 5-HT in infection via AhR. Thus, because 5-HT activates AhR[22], the positive loop between AhR and 5-HT likely fulfills the service of 5-HT response to exposome[52].

The importance of the Trp-derived serotonin-kynurenine balance for microbial and immune homeostasis is increasingly being appre-ciated in both the gut[53] and the lung[19,54]. However, the contribution of the three metabolic pathways to Trp partitioning in the lung is less known. We found here that the three Trp metabolic pathways are coordinately activated in fungal pneumonia in the condition of 5-HT sufficiency but not deficiency, the last condition being associated with the defective IDO1 and unrestrained AhR activity. This suggests that the Tph1/5-HT, by sustaining the IDO1 and restraining AhR activation, exerts a critical control upon the dynamic activation of the two path-ways in infection. We have indeed shown that the optimal antifungal resistance and immune tolerance in infection depends on the timely regulated expression of either pathway[24, 25]. The microbiota and metabolites pivotally contributed to this coordination activity of 5-HT. Indeed, the network topology of enzymatic association network revealed that Trp-metabolizing bacteria, the associated enzymes along the indole pathways, and a number of indole derivatives, all known AhR-agonists[55], were highly expanded, expressed, and produced in 5-HT-deficient mice, these finding being consistent with the unrest-rained AhR activation in these mice. In contrast, the relative absence of metabolites known to activate IDO1, namely NAS or 5-OH-IAA, likely contributed to the defective IDO1 activation in these mice. Whether the defective IDO1 activation or the exuberant AhR activity or both contribute to the inflammatory tone in $Tph1^{-/-}$ mice is hard to tell at this point. As a matter of fact, both the pathogenic role of defective host IDO1/Kyn pathway[24, 56] and the beneficial role of the indole/AhR pathway[25,57] in lung diseases are well-known. It is tempting to speculate that unrestrained activation of microbial-dependent AhR in the rela-tive absence of host Tph1-IDO1 braking pathways may lead to unre-solved inflammation. Alternatively, consistent with the promiscuous activity of indole ligands capable of activating additional xenobiotic

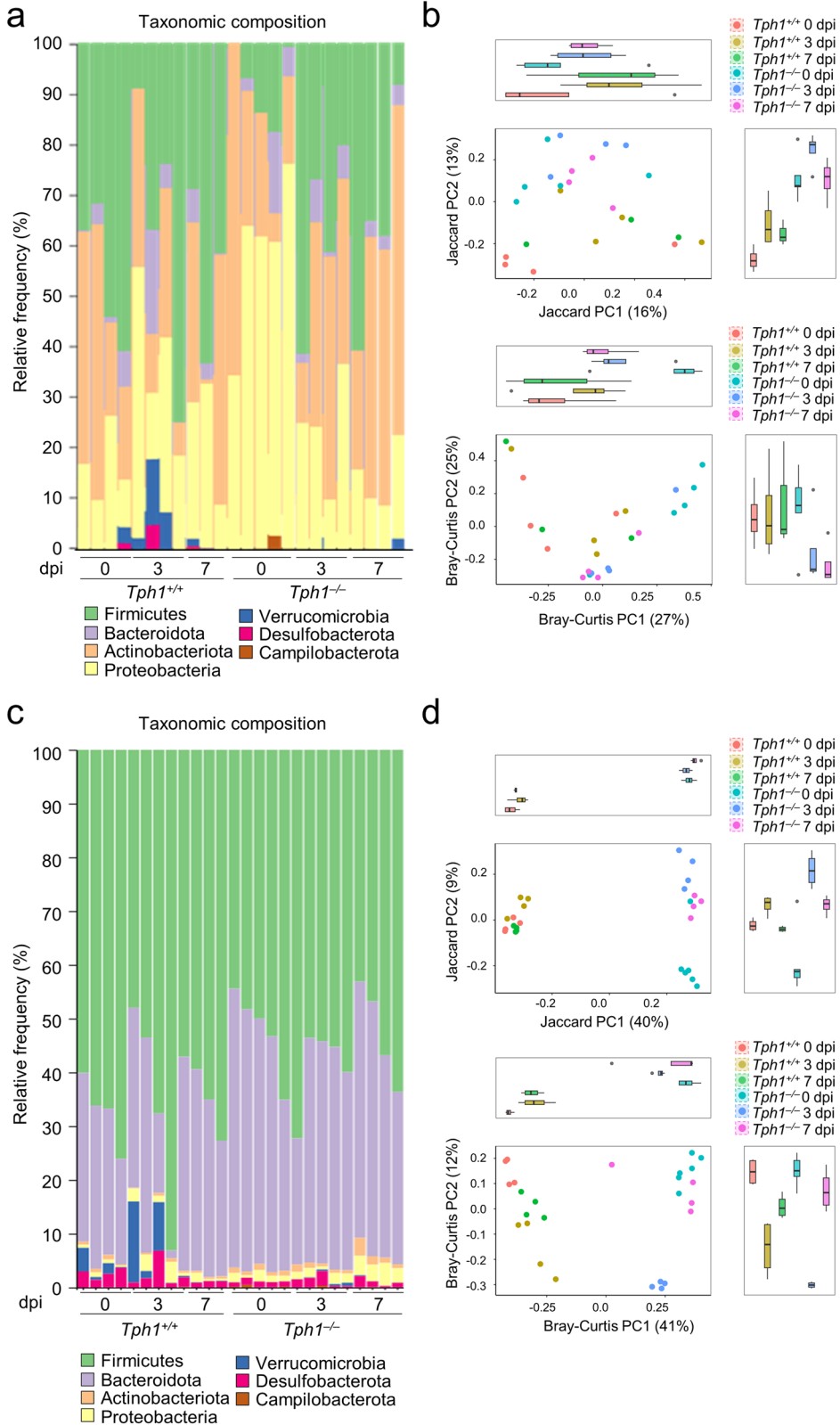

**Fig. 5 | 5-HT deficiency affects lung and gut microbiota composition.** Microbial composition in the lung (**a**, **b**) (*n* = 3–5 mice/group) and gut (**c**, **d**) (*n* = 4–6 mice/group) of *A. fumigatus*-infected *Tph1*⁺/⁺ and *Tph1*⁻/⁻ littermates at 0, 3, and 7 days post infection (dpi). Barplot showing lung (**a**) and gut (**c**) bacterial composition (abundance percentage) of each sample at phylum level. Taxa are differentiated by colors. PCoA of beta diversities (Jaccard and Bray-Curtis) indexes in lung (**b**) and gut

(**d**) at the sequence variant level. Scatterplots of the first two components are represented together to the marginal boxplots (pairwise Wilcoxon test results are shown in Table S3). **b**, **d** Whiskers represent the minimum and maximum (unless points extend 1.5 * IQR from the hinge, then shown as individual points), the box represents the interquartile range, and the center line represents the median.

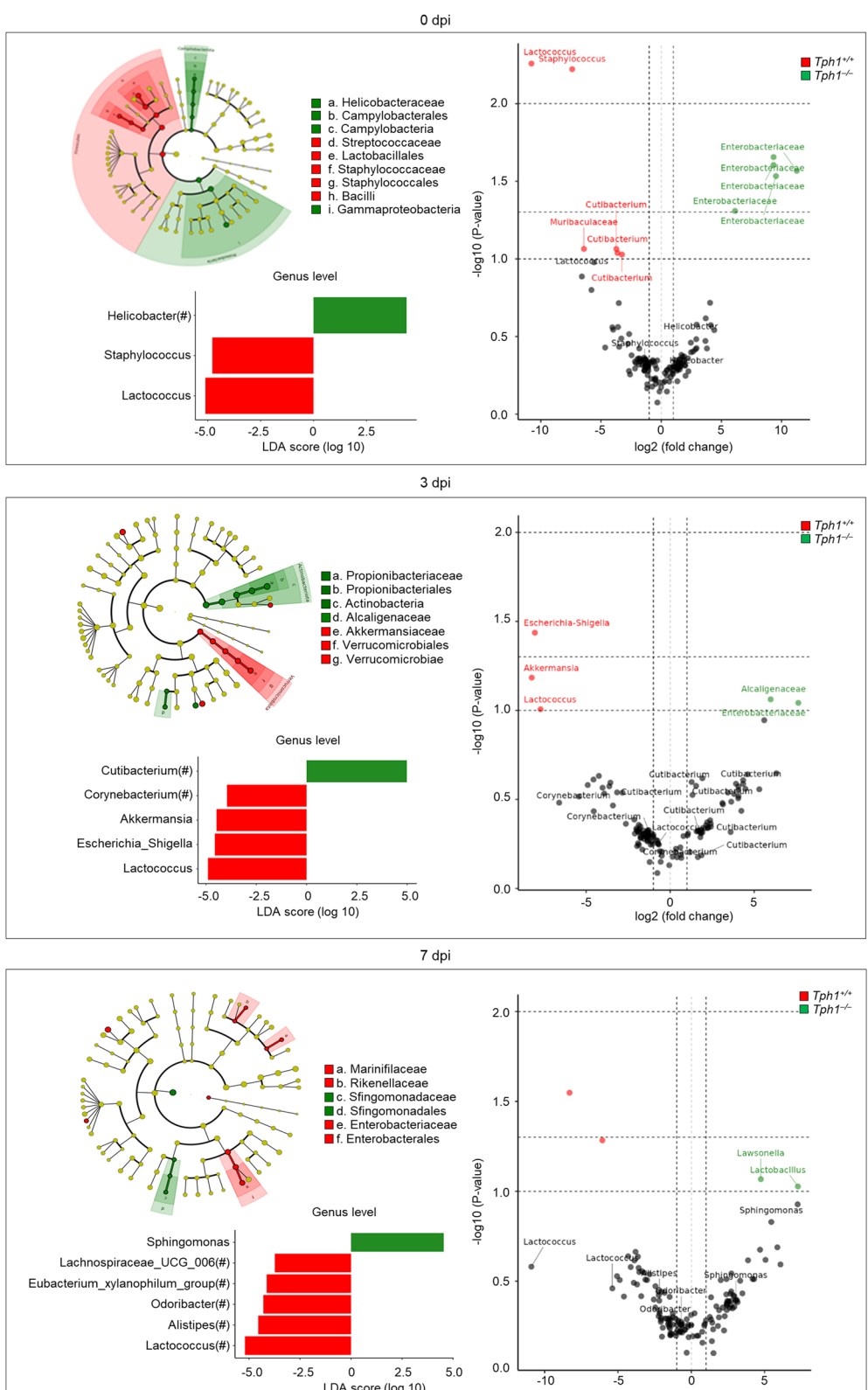

receptors, such as the pregnane X receptor[58], it is possible that other receptors and signaling pathways are at work in *Tph1*[−/−] mice. In this regard, it is worth mentioning that the microbial arginine-putrescin biosynthesis pathway, known to be AhR-driven[59] and dysregulated in several lung diseases including CF[60], was increased in *Tph1*[−/−] mice. Lastly, the finding that kynurenines and downstream metabolites were produced in the relative absence of host IDO1, clearly points to the

metabolic activity of the kynurenine-producing Firmicutes, Bacteriodota and Proteobacteria[61]. However, as kynurenines are known to mediate immunosuppression via AhR, this raises the intriguing observation of divergent signaling between microbial and host kynurenines, an observation supported by the diversity of bacterial and eukaryotic kynureninase[62] and by the influence of microbes expressing the kynurenine pathway enzymes on the host's

**Fig. 6 | 5-HT deficiency associates with distinct microbial profiles in the lung during infection.** Taxonomic visualization of statistically and biologically consistent differences between *Tph1⁺/⁺* and *Tph1⁻/⁻* littermates at 0 (upper panels), 3 (middle panels) and 7 (bottom panels) days post infection (dpi) by LEfSe (cladograms and histograms, left panels) and Q2-ALDEx2 (volcano plots, right panels) at genus and sequence variant levels, respectively. Genera (circles) colored green or red in cladograms are significantly associated with the group indicated by the corresponding legend. Circles colored in yellow are not significantly associated with either group. The size of each circle is proportional to the abundance of the corresponding genus in the dataset. Boxplots of the genera that LEfSe significantly associates to either *Tph1⁺/⁺* and *Tph1⁻/⁻* mice are reported under the relative cladograms. Volcano plots show association results of the Q2-ALDEx2 analysis according to the color legend (taxonomies of genera present in the LEfSe results are emphasized). Horizontal dashed gray lines represent *p* values equal to 0.1, 0.05, and 0.01. Vertical dashed gray lines represent a fold-change between groups of interest equal to 2. LDA linear discriminant analysis, LEfSe linear discriminant analysis together with effect size evaluation.

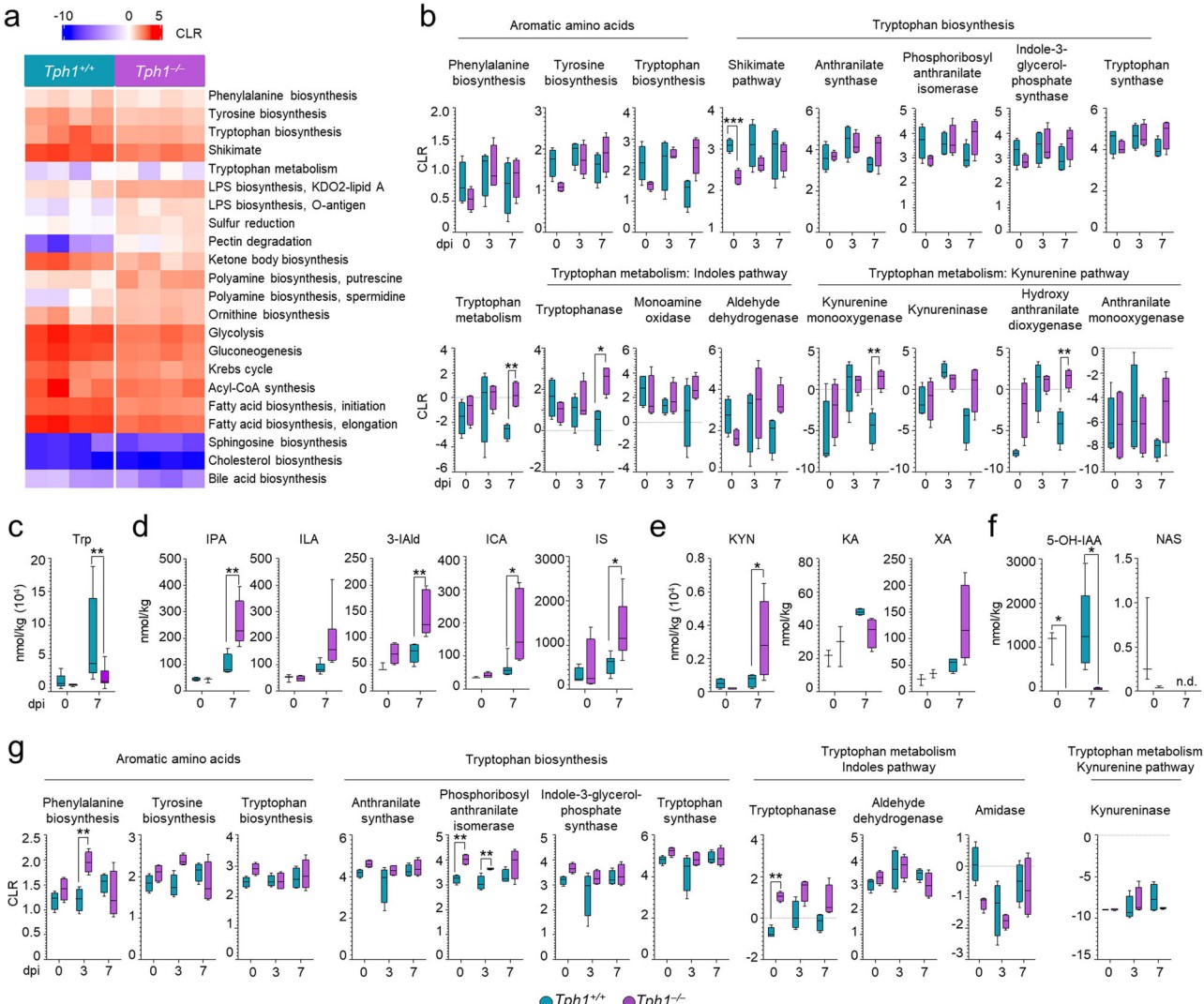

**Fig. 7 | 5-HT deficiency affects metabolite profile in the lung and feces.**
**a**–**f** Metabolite production in the lung of *A. fumigatus*-infected *Tph1⁺/⁺* and *Tph1⁻/⁻* littermates at 0, 3, and 7 days post infection (dpi). **a** Heatmap representing metabolic functions inferred by PICRUSt2 analysis (KEGG functions) associated with *Tph1⁺/⁺* or *Tph1⁻/⁻* uninfected mice. **b** Boxplots of metabolic functions and enzyme inferred by PICRUSt2 analysis (KEGG function and enzyme) (*n* = 4 mice/group). Levels of **c** Trp (*n* = 5 mice/group in 0 dpi and *n* = 13 mice/group in 7 dpi), **d** Indole-3-propionic acid (IPA), Indole-3-lactic acid (ILA), Indole-3-carboxaldehyde (3-IAld), Indole-3-carboxylic acid (ICA), Indoxyl sulfate (IS) (*n* = 3–4 mice/group in 0 dpi and *n* = 6–7 mice/group in 7 dpi), **e** Kynurenine (KYN), Kynurenic acid (KA), xanthurenic acid (XA) (*n* = 3–4 mice/group in 0 dpi and *n* = 4–5 mice/group in 7 dpi), **f** 5-Hydroxyindole 3-Acetic Acid (5-OH-IAA) and N-Acetylserotonin (NAS) measured in the lung homogenates by mass spectrometry (*n* = 3 mice/group in 0 dpi and *n* = 5 mice/group in 7 dpi). **g** Boxplots of aromatic amino acids and tryptophan metabolic functions and enzyme inferred by PICRUSt2 analysis (KEGG function and enzyme) in the feces (*n* = 4 mice/group). Data are expressed as mean ± SEM and representative from two to four experiments. *\*p* < 0.05. *Tph1⁻/⁻ vs Tph1⁺/⁺* mice. Multiple *t* tests. Statistical significance was determined by Holm−Sidak method, with alpha = 0.05. CLR, centered log.ratio. Adjusted *P* values for **b**: 0.0005 (Shikimate pathway), 0.0030 (Tryptophan metabolism), 0.0081 (Tryptophanase), 0.0075 (Kynurenine monooxygenase), 0.0082 (Hydroxyanthranilate dioxygenase); for **c**: 0.0089; for **d**: 0.0013 (IPA), 0.0063 (3-IAld), 0.0311 (ICA), 0.0246 (IS); for **e**: 0.0485; for **f**: for 5-OH-IAA, 0.0137 (0 dpi), 0.0137 (7 dpi); for **g**: 0.0091 (Phenylalanine biosynthesis), for Phosphoribosyl anthranilate isomerase, 0.0025 (0 dpi), 0.0096 (3 dpi), 0.0001 (Tryptophanase). **b**, **g** Whiskers represent the minimum and maximum, the box represents the interquartile range, and the center line represents the median. Source Data are provided as a Source Data file.

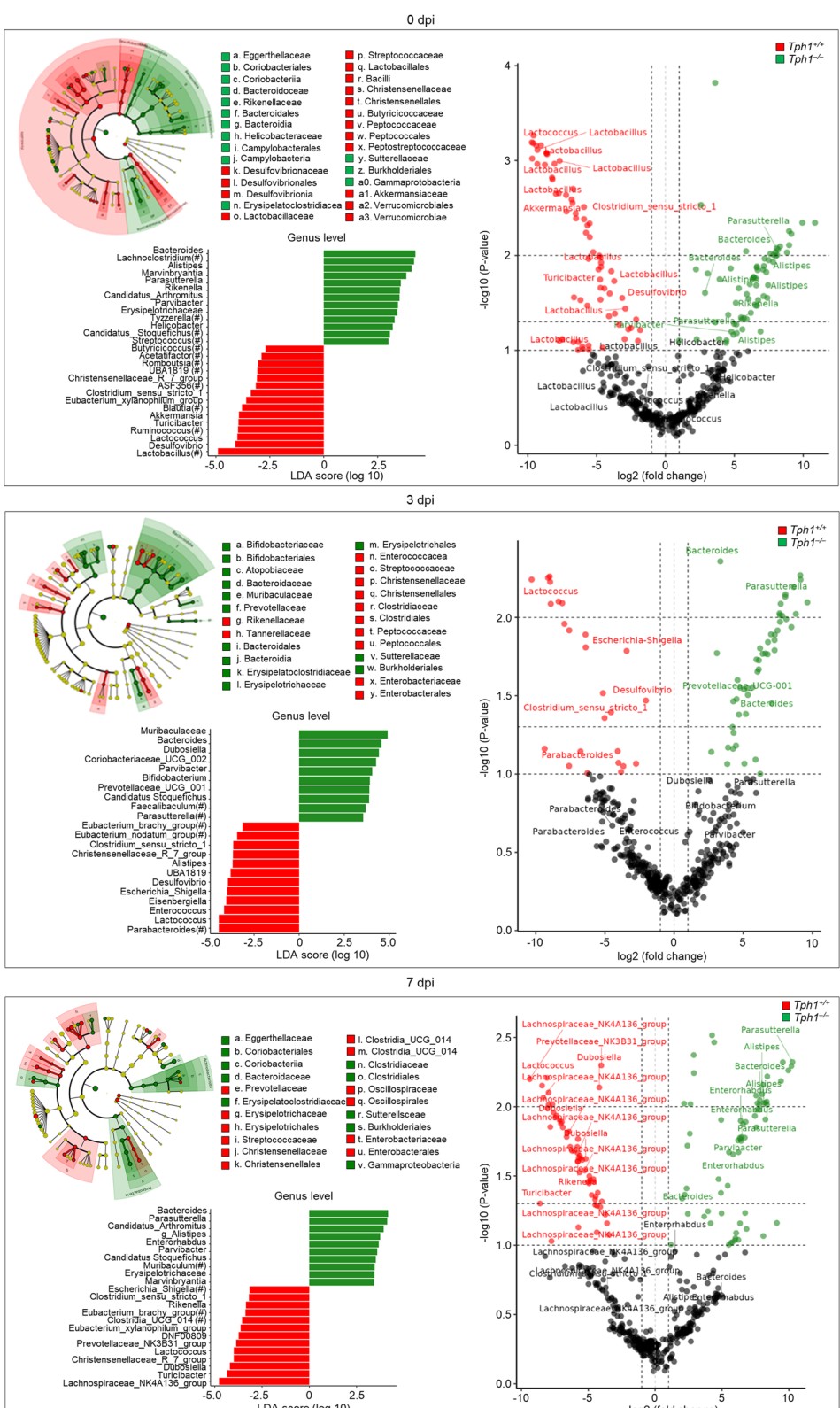

immunometabolic activity[63, 64]. In this regard, our preliminary evidence are of a distinct AhR signaling pathway occurring in *Tph1⁻/⁻* mice that includes the AhR-target gene *Serpinb2*, as suggested,[65, 66] that is up-regulated under multiple inflammatory conditions, and dysregulated expression and polymorphisms are associated with several human inflammatory diseases[67], including type 2 airway inflammation[68].

Considering the existence of a bidirectional gut-lung axis involving host-microbe as well as microbe-microbe-interaction[43, 69, 70], it was not surprising to find that the gut microbiota had an impact on fungal pneumonia. Indeed, Trp-metabolizing bacteria were also expanded in the gut of 5-HT-deficient mice and likely contributed to the production of indoxyl sulfate that results from the sulfonation of bacterially-derived indole via sulfotransferases in liver[71]. As a matter of fact,

**Fig. 8 | 5-HT deficiency associates with distinct microbial profiles in the feces during infection.** Taxonomic visualization of statistically and biologically consistent differences between *Tph1*[+/+] and *Tph1*[−/−] littermates at 0 (upper panels), 3 (middle panels) and 7 (bottom panels) days post infection (dpi) by LEfSe (cladograms and histograms, left panels) and Q2-ALDEx2 (volcano plots, right panels) at genus and sequence variant level, respectively. Genera (circles) colored green or red in cladograms are significantly associated with the group indicated by the corresponding legend. Circles colored in yellow are not significantly associated with either group. The size of each circle is proportional to the abundance of the

corresponding genus in the dataset. Boxplots of the genera that LEfSe significantly associates with either *Tph1*[+/+] and *Tph1*[−/−] mice are reported under the relative cladograms. Volcano plots shows association results of the Q2- ALDEx2 analysis according to the color legend (taxonomies of genera present in the LEfSe results are emphasized). Horizontal dashed gray lines represent *p* values equal to 0.1, 0.05, and 0.01. Vertical dashed gray lines represent a fold-change between groups of interest equal to 2. LDA linear discriminant analysis, LEfSe linear discriminant analysis together with effect size evaluation.

transplantation of *Tph1*[−/−] fecal microbiota increased AhR activity in the lung. However, the relative absence of *Verrucomicrobia* in *Tph1*[−/−] feces could also be working, given the protective role of gut *A. muciniphila* and *Actinobacteria* spp. in lung diseases. Whatever the case, the effects of fungal pneumonia on gut microbiota were also an interesting observation. While the distal effects of viral lung infections on the intestinal microbiota are known[72], the evidence linking the lung infection to the intestinal manifestations occurring in patients with fungal pneumonia are, to our knowledge, poorly, but worth to be, understood.

Despite some limitations of the present study, inherently associated with the use of low-biomass samples and therefore requiring a deep control of contaminants potentially affecting the taxonomic composition of low-biomass sample and a more robust group sizes for a meaningful statistical comparison, our results may have clinical implication in CF. While advances in health care have prolonged lifespan, repeated hospitalization and management of a chronic disease contributes to poor mental health in CF patients including depression and anxiety[73]. In contrast to an earlier study showing no decreased blood 5-HT level in children with CF[74], we found here that 5-HT levels were lower in the sputa of patients with CF likely due to intense 5-HT catabolism as revealed by the increased levels of 5-OH-IAA in both mice (this study) and humans[75] with CF. Considering that SSRIs are recommended as first-line medications for both depression and anxiety in most adolescents and adults with CF and that pharmacokinetics and therapeutic activity of SSRIs may be altered in CF by concurrent therapy[76], our study suggests that the therapeutic drug monitoring of 5-HT may also be.

# Methods

## Ethics statement
Mouse experiments were performed according to Italian Approved Animal Welfare Authorization 360/2015-PR and Legislative Decree 26/2014 regarding the animal license obtained by the Italian Ministry of Health lasting for 5 years (2015–2020) and the Welfare Authorizations 662/2020-PR, lasting for two years (2020–2022) and Legislative decree 26/2014 regarding the animal license, obtained by the Italian Ministry of Health. Human studies approval was obtained from institutional review boards the Bambino Gesù Children's Hospital (Rome, Italy), Ospedale Maggiore Policlinico, University of Milan (Milan, Italy), Innsbruck Medical University (Innsbruck, Austria) and Servizio di Supporto Fibrosi Cistica (Cerignola, Foggia, Italy). Informed consent was obtained from all participants.

## Cell culture and treatment
Mouse hepatoma H1L6.1c3 (containing pGudLuc6.1) cells were maintained in alpha-minimum essential media (α-MEM, Gibco), supplemented with 10% (v/v) fetal bovine serum (FBS) and 1% (v/v) penicillin/streptomycin solution. Cells were stimulated with lung homogenates of infected or not mice for 24 h and with 10 and 100 μM of 3-IAld, 5-OH-IAA, and 5-HT (Sigma-Aldrich) for the same time. Alveolar macrophages from the lung of *Tph1*[+/+] and *Tph1*[−/−] uninfected mice were obtained after 2 h of plastic adherence at 37 °C, were grown in RPMI medium supplemented with 10% (v/v) FBS and pulsed with *A. fumigatus* conidia.

## Mice
C57BL/6 mice were purchased from Charles River Laboratories (Calco, Italy). *Tph1*[+/+], *Tph1*[−/−] from Charles River Laboratories (Calco, Italy). C57BL6-*Kit*[W/W-v77] and B6.129-Ahrtm1Bra/J Ahr-deficient (*Ahr*[−/−])[25] were bred under specific pathogen-free conditions in the Animal Facility of Perugia (Perugia, Italy). CF mice are homozygous for the Phe508del-*Cftr* (*Cftr*tm1EUR, F508del) allele[77]. Six-to 8-week-old male and female mice weighing 20–25 g were used in all experiments. Mice were fed a normal chow (1310 formula, Altromin) and housed at a 12 light/12 dark cycle, with a temperature of 18–23 °C and a 40–60% humidity.

## Phagocytosis assay
To assess phagocytosis, adherent macrophages ($2 \times 10^5$/well in 24-well plates) purified from lungs of uninfected mice and thioglycollate-elicited peritoneal neutrophils were pulsed with *A. fumigatus* conidia at a 1:3 effector-to-target ratio. Cells were incubated for 90 min a 37 °C at 5% $CO_2$. The samples was collected and centrifuged at $1800 \times g$ at 4 °C for 5 min. For cell counts, cytospin preparations were stained with May-Grünwald Giemsa reagents. The number of cells or adherent macrophages with ingested conidia was enumerated by examining the slides using a microscope BX51 microscope (Olympus), and data were expressed as percentage of cells or adherent macrophages that internalized one or more conidia.

## Measurement of reactive oxygen species production
Phagocytic cells from uninfected mice were plated ($1 \times 10^4$/well) in 96-well dark clear bottom plates (Corning) and pulsed with *A. fumigatus* conidia at a 1:1 effector-to-target ratio. Infection were synchronized by centrifugation at $100 \times g$ for 5 min, and then 50 μM dihydroethidium (DHE) and 10 μM Dihydrorhodamine 123 (DHR) (Sigma-Aldrich), was added to each well, and the production of reactive oxygen species was measured for 1 h period using multifunctional microplate reader Tecan Infinite 200 (Tecan). Excitation was performed at a wavelength of 518 nm and emission was measured at a wavelength of 605 nm.

## Infection, allergy and treatments
For pulmonary infection, viable conidia from *A. fumigatus* Af293 strain were obtained as described[78]. Mice were anesthetized in a plastic cage by inhalation of 3% isoflurane (Forane, Abbott) in oxygen before the intranasal instilling of $6 \times 10^7$ (or $10^8$ in selected experiments) resting conidia/20 μL of saline. Mice were treated intraperitoneally with 5.5 mg/mouse of Parachlorophenylalanine (PCPA) or 5-Hydroxytryptophan (5-HTP) one day before infection and every other days[29]. Indoximod (2 mg/mouse) (Selleckchem) or CH-223191 (0.2 mg/mouse) (Sigma-Aldrich) were administered intraperitoneally every other days after the infection. Inhibition of the IDO1 or the AhR pathway was observed upon the respective treatment as reported[79, 80]. Mice were sacrificed at different time points. For allergic bronchopulmonary aspergillosis (ABPA), mice were sensitized by concomitant intraperitoneal and subcutaneous administration (100 μg) of *A. fumigatus* culture filtrate extract (cell culture filtrate antigens, CCFA) dissolved in incomplete Freund's adjuvant (Sigma-Aldrich, USA). Twice at weekly interval, 20 μg of the extract was instilled to these mice by intranasal route. At third week, mice were infected intranasally with *A. fumigatus* and sacrificed 7 days post infection (dpi). For vaginal

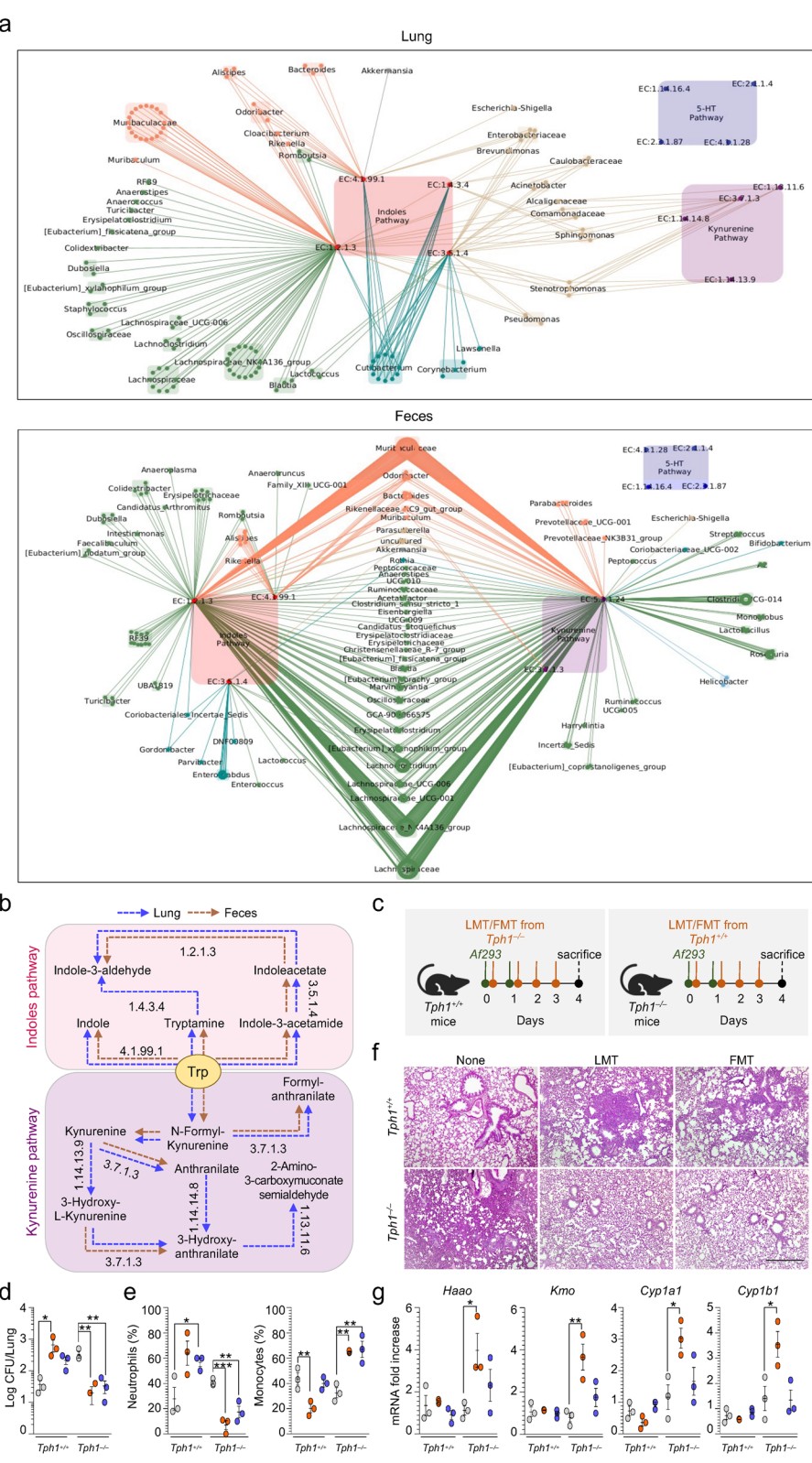

infection, estrogen-treated mice were inoculated intravaginally with $5 \times 10^6$ viable C *andida albicans* 3153 A blastospores from early-stationary-phase cultures. In fungal pneumonia, mice were evaluated for survival (%) or sacrificed at different time points and evaluated for inflammatory parameters. Fungal growth, expressed as colony-forming units ($\log_{10}$ CFU) per organs, was obtained by serially diluting homogenates on Sabouraud agar plates incubated at 37 °C for 24 h.

In vaginal infection, CFU were enumerated after incubation of vaginal fluids.

### Collection of bronco-alveolar lavage

Lungs were filled thoroughly with 2.0 ml aliquots of pyrogen-free saline through a 22-gauge bead-tipped feeding needle introduced into the trachea. Bronco-alveolar lavage (BAL) fluid was collected in a

**Fig. 9 | The local and distal microbiota contribute to the IDO1/AhR axis in the lung. a** Cytoscape network of lung and gut microbiome that qualitatively contributes to Indole (red), Kynurenine (purple), and 5-HT (cyan) pathways. Taxa identified by Q2- ALDEx2 as driving the difference between $Tph1^{-/-}$ and $Tph1^{+/+}$ and contributing with an effect size versus $Tph1^{-/-}$ littermates at dpi 7 were used to create networks. Each hexagon represents a bacterial sequence variant, each dot represents an enzyme in a pathway, according to the dot label. Taxa are grouped by genus, explicitly shown in the figure, and colored by phylum (Actinobacteriota, darkcyan; Bacteroidota, salmon1; Firmicutes, palegreen; Verrucomicrobiota, honeydew; Proteobacteria, tan; Campilobacterota, darkgrey). Edges show the qualitative contribution (presence) of each taxon to enzymes, and all have the same width. **b** Description of tryptophan catabolic pathways in lung (blue arrows) and feces (brown arrows) of $Tph1^{-/-}$ mice. **c**–**g** *A. fumigatus*-infected $Tph1^{+/+}$ and $Tph1^{-/-}$ mice were transplanted with fresh lung homogenates or fecal pellets from uninfected $Tph1^{+/+}$ or $Tph1^{-/-}$ mice as depicted in the experimental schedule (**c**). Mice were sacrificed at 4 days post infection and assessed for **d** fungal burden, **e** % neutrophils

and monocytes in the bronco-alveolar lavage, **f** lung histology (periodic acid-Schiff staining) and **g** gene expression in the lungs (RT-PCR). Photographs were taken with a high-resolution microscope (Olympus BX51), ×10 magnification (scale bar 500 μm). Data are expressed as mean ± SEM and representative from two experiments. *$p < 0.05$, **$p < 0.01$, and ***$p < 0.001$, LMT or FMT vs untreated (None) mice. The in vivo groups consisted of three mice/group. LMT, lung microbiota transplantation. FMT, fecal microbiota transplantation. One-way ANOVA, Dunnett post hoc test. Adjusted P values for **d**: 0.0354 ($Tph1^{+/+}$, None vs LMT), 0.0068 ($Tph1^{-/-}$, None vs LMT), 0.0083 ($Tph1^{-/-}$, None vs FMT); for **e**: 0.0495 (Neutrophils (%), $Tph1^{+/+}$, None vs FMT), 0.0008 (Neutrophils (%), $Tph1^{-/-}$, None vs LMT), 0.0054 (Neutrophils (%), $Tph1^{-/-}$, None vs FMT), 0.0084 (Monocytes (%), $Tph1^{+/+}$, None vs LMT), 0.0033 (Monocytes (%), $Tph1^{-/-}$, None vs LMT), 0.0025 (Monocytes (%), $Tph1^{-/-}$, None vs FMT); for **g**: 0.0368 (Haao, $Tph1^{-/-}$, None vs LMT), 0.0074 (Kmo, $Tph1^{-/-}$, None vs LMT), 0.0248 (Cyp1a1, $Tph1^{-/-}$, None vs LMT), 0.0301 (Cyp1b1, $Tph1^{-/-}$, None vs LMT). Source Data are provided as a Source Data file.

plastic tube on ice and centrifuged at $1800 \times g$ at 4 °C for 5 min. For differential BAL fluid cell counts, cytospin preparations were stained with May-Grünwald Giemsa reagents. At least 10 fields (200 cells/field) were counted, and the percent of neutrophils was calculated. Photographs were taken using a high-resolution Olympus DP71 camera (Olympus BX51).

### Histology
The tissues were removed and fixed in 10% phosphate-buffered formalin (Bio Optica), embedded in paraffin, and sectioned at 3 μm. For histology, paraffin-embedded tissues were stained with Periodic Acid-Schiff (PAS) and Gomori methenamine-silver staining. Histology sections were observed using a BX51 microscope (Olympus), and images were captured using a high-resolution DP71 camera (Olympus). The degree of microscopic injury was scored based on the extent of infiltration of inflammatory cells (perivascular, peribronchial, parenchymal), following the classification: none (score 0), slight (score 1), moderate (score 2), marked (score 3), and severe (score 4).

### Immunofluorescence
For immunofluorescence, the perfused lung sections were rehydrated and, after antigen retrieval in citrate buffer (10 mM, pH 6), fixed in 4% formaldehyde (ChemCruz) for 20 min at room temperature and permeabilized in a blocking buffer containing 5% BSA. The slides were then incubated at 4 °C with primary antibodies anti-serotonin (clone 5HT-H209, NOVUS Biologicals, NB120-16007V2), anti-Pgp9.5 (clone EPR4118, Abcam, ab108986), anti-mast cell chymase (Bioss, bs-2353R), anti-IDO1 (Millipore, AB9900), anti-AhR (clone RPT1, Invitrogen, AB_2273723) and anti-CD41 (Emfret, M025-1). After extensive washing with phosphate-buffered saline (PBS), the slides were then incubated with secondary antibody goat anti-Rabbit Alexa Fluor 488 (Abcam) (for Pgp9.5, chymase, IDO1) or goat anti-Mouse Alexa Fluor 555 (Abcam) (for 5-HT and AhR). Nuclei were counterstained with Hoechst 33342 (Invitrogen, Molecular Probes). Images were acquired using a BX51 microscope with a high-resolution DP71 camera (Olympus). Average intensities were calculated by using Image J software calculated on $142 \times 142$ pixel area[81].

### TUNEL assay
Terminal deoxynucleotidyl transferase dUTP nick-end-labeling (TUNEL) staining was performed using the In Situ Cell Death Detection Kit, POD (Roche Diagnostics, Germany) according to the manufacturer's instructions. The sections were mounted and analyzed by fluorescent microscopy and TUNEL-positive cells were quantitated as percentage of cells positive from five high-powered fields in each section.

### RT-PCR
Real-time PCR was performed using the CFX96 Touch Real-Time PCR detection system and iTaq Universal SYBR Green Supermix (Bio-Rad). Lung or thoracic lymph nodes were lysed and total RNA was isolated with TRIZOL Reagent (Thermo Fisher Scientific) and cDNA was synthesized using the PrimeScript RT Reagent Kit with gDNA Eraser (Takara), according to the manufacturer's instructions. Amplification efficiencies were validated and normalized against β-*actin*. The thermal profile for SYBR Green RT-PCR was at 95 °C for 3 min, followed by 40 cycles of denaturation for 30 s at 95 °C, and an annealing/extension step of 30 s at 60 °C. Each data point was examined for integrity by analysis of the amplification plot. The murine primers used in this study (ThermoFisher) are detailed in Supplementary Materials (Table S5).

### ELISA
Cytokine content was determined in lung homogenates by using specific ELISA kits according to the manufacturers' instructions (Invitrogen, Biolegend, eBioscience Inc., Life Technologies and R&D System, LDN). The concentration of secreted cytokines was expressed as pg of cytokine for mL. 5-HT was analyzed in lung homogenates or serum by using specific kits (LDN).

### Sputum analysis
Spontaneous sputum from patients with the *ΔF508* homozygous mutation[77] was collected in sterile cup and immediately processed. The sputum was washed with NaCl 150 mM, mixed with an equal volume (1:1) of Sputasol®, and then incubated in water bath at 37 °C for 15 min until visible homogenous. Processed sputum was centrifuged, and supernatant was collected for 5-HT evaluation with specific kit (LDN).

### Multiplex cytokine/chemokine immunoassay
Mouse cytokine and chemokine were measured in lung homogenates using the Cytokine & Chemokine 26-Plex Mouse ProcartaPlex™ Panel 1 (Life Technologies), as indicated in the manufacturer's protocol. Plates were read using Magpix (Luminex). Data analysis was performed by Procartaplex Analysis App.

### Western blot analysis
For western blot, the lung was homogenate in RIPA buffer (Tris/HCl at pH 8.0, 50 mM, NaCl 150 mM, SDS 0.1%, sodium deoxycholate 1%, Triton X-100 1%), supplemented with protease inhibitor cocktail (PIC, Roche) and phenylmethylsulfonyl fluoride (PMSF, Sigma-Aldrich). The proteins were separated by electrophoresis on SDS-PAGE and detected using specific antibodies against AhR (Proteintech, 17840-1-AP) and IDO1 (Millipore,

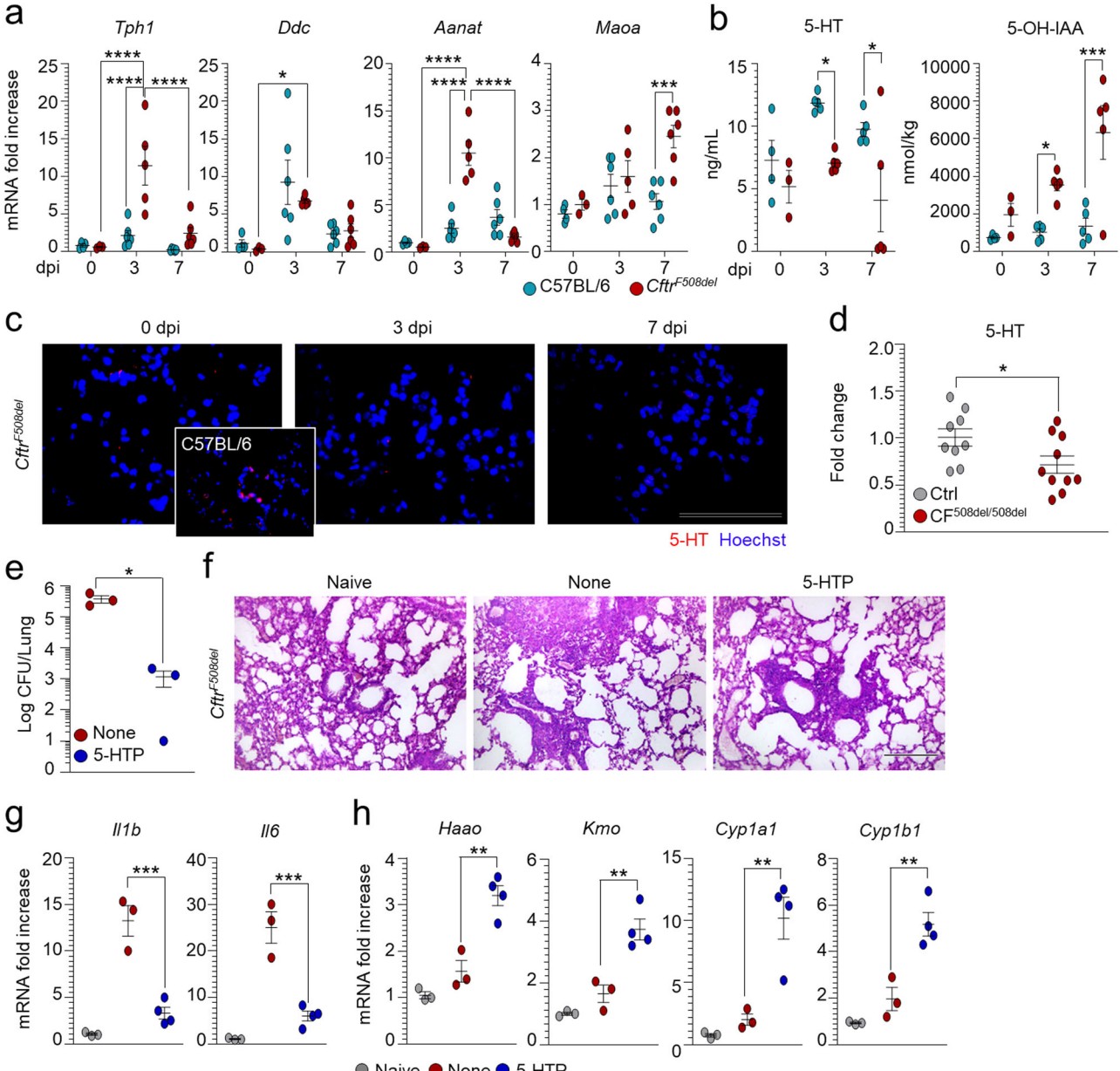

**Fig. 10 | 5-HT is defective in cystic fibrosis.** C57BL/6 and *Cftr^F508del* mice were infected intranasally with *A. fumigatus* and assessed at 3 or 7 days post infection (dpi) for **a** *Tph1, Ddc, Aanat, Maoa* expression in lungs (RT-PCR) (*n* = 3–4 mice/group at 0 dpi, *n* = 5–6 at 3 and 7 dpi), **b** levels of 5-HT and 5-OH-IAA in lungs homogenates (*n* = 3–4 mice/group at 0 dpi, *n* = 5 at 3 and 7 dpi), and **c** 5-HT immunofluorescence staining in perfused lung. **d** Mean fold-change normalization of 5-HT levels in sputa of patients with the *ΔF508* homozygous mutation (*n* = 10) and control (Ctrl) subjects (*n* = 9). **e**–**h** *A. fumigatus*-infected *Cftr^F508del* mice were treated with 5-HTP as in legend to Fig. 2 and assessed at 7 dpi for **e** fungal growth (*n* = 3 mice/group), **f** lung histology (periodic acid-Schiff staining) and **g, h** gene expression in the lung (RT-PCR) (*n* = 3 mice/group in Naive and None, *n* = 4 mice/group in 5-HTP). Hoechst was used for nuclear counterstain in blue. Photographs were taken with a high-resolution microscope (Olympus BX51), ×20 magnification (scale bar 200 μm). Data are expressed as mean ± SEM and representative from two experiments. *\*p < 0.05, \*\*p < 0.01, \*\*\*p < 0.001, and \*\*\*\*p < 0.0001, Cftr^F508del vs* C57BL/6 mice, infected *vs* uninfected mice, CF^508del/508del patients vs controls. **a** Two-way ANOVA, Tukey post hoc test. Adjusted *P* values: for *Tph1,* <0.0001 (*Cftr^F508del*, 0 vs 3 dpi), <0.0001 (*Cftr^F508del*, 3 vs 7 dpi), <0.0001 (3 dpi, C57BL/6 vs *Cftr^F508del*); for *Ddc,* 0.050 (*Cftr^F508del*, 0 vs 3 dpi); for *Aanat,* <0.0001 (*Cftr^F508del*, 0 vs 3 dpi), <0.0001 (*Cftr^F508del*, 3 vs 7 dpi), <0.0001 (3 dpi, C57BL/6 vs *Cftr^F508del*); for *Maoa,* 0.0005 (7 dpi, C57BL/6 vs *Cftr^F508del*). **b** Two-way ANOVA, Sidak post hoc test. Adjusted *P* values: for 5-HT, 0.0492 (3 dpi, C57BL/6 vs *Cftr^F508del*), 0.0174 (7 dpi, C57BL/6 vs *Cftr^F508del*); for 5-OH-IAA, 0.0494 (3 dpi, C57BL/6 vs *Cftr^F508del*), 0.0001 (7 dpi, C57BL/6 vs *Cftr^F508del*). **d** Mann Whitney test, two-tailed, *P* value 0.0279. **e** Unpaired t-test, two-tailed, *P* value 0.0199. **g, h** One-way ANOVA, Bonferroni post hoc test, Adjusted *P* values for **g:** 0.0004 (*Il1b*), 0.0005 (*Il6*); for **h:** 0.0018 (*Haao*), 0.0034 (*Kmo*), 0.0063 (*Cyp1a1*), 0.0037 (*Cyp1b1*). Source Data are provided as a Source Data file.

AB9900). Normalization was performed with α-actin (Sigma-Aldrich, A2066), β-tubulin (clone D-10, Santa Cruz Biotechnology, sc-5274), or Gapdh (clone 6C5, Santa Cruz Biotechnology, sc-32233). Protein signal intensities were quantified by densitometric analysis using Image J software.

## Sample collection, processing, and sequencing for microbial composition

DNA was isolated from murine feces by the QIAamp PowerFecal Pro DNA kit (Qiagen) according to the manufacturer's instructions. In the case of lung tissues, the following modification was introduced.

Specifically, 20 mg of tissue was added to PowerBead tubes 1.4 mm (Qiagen) and resuspended in 800 μl of CD1 solution. Mechanical lysis was performed with a bead-beater (BIO101/Savant FastPrep FP120) at 4.0 m/sec for 20 sec, followed by 20 sec in ice. The mechanical lysis was repeated twice before centrifugation and transfer to the Power-Bead Pro Tubes. The bacterial microbiota was evaluated by 16S rRNA sequencing performed at Bio-Fab Research (next-generation sequencing; Rome, Italy). The V4–V5 hypervariable regions of the bacterial 16S rRNA gene were amplified by the 515YF/926R (GTGY-CAGCMGCCGCGGTAA/CCGYCAATTYMTTTRAGTT) primer pair, and amplicons were sequenced on a 300 bp paired-end read sequencing on the Illumina MiSeq platform (V3).

## Microbiome analysis

Fastq format of demultiplexed libraries obtained from Bio-Fab company data were subjected to a quality control procedure by using FASTQC (version 0.11.9) software[82]. Illumina TruSeq adapters were removed from high-throughput sequencing data by applying Cutadapt (version 2.8) command line to all sequenced files[83]. In particular, reads with at least 100 nucleotides (nt) were retained, a low-quality cut-off equal to 5 was used to trim the 3′ end of each read and a maximum error rate on searches for adapter sequences was set to 5%. Next, Cutadapt was also used for detecting and removing forward and reverse primer-sequences with an error rate equal to 10% in both fastq files R1 and R2. Primer-clipped sequences were then oriented into forward-reverse primer direction by saving as R1.fastq and as R2.fastq files the reads where forward and reverse primers, respectively, were detected. These paired-end sequences were imported in the next-generation microbiome bioinformatics platform Qiime2 platform (version 2022.2)[84] embedded in our genomic cloud-computing environment using the built-in command ‘qiime tool import’. Thus, paired-end sequences were denoised, dereplicated, filtered by both any phiX reads and chimera (consensus), by using q2-dada2 quality control method[85] implemented in the ‘qiime dada2 denoise-paired’ Qiime2 plugin. In particular, the q2-dada2 method retains and join reads with a minimum overlap of 12 nt and makes use of sequence error profiles to obtain putative error-free sequences, referred to as either sequence variants (SVs) or 100% operational taxonomic units (OTUs). The q2-dada2 plugin parameters have been set in order to truncate forward reads at position 280 while guaranteeing no more than 5 incorrected bases and to truncate reverse reads at position 150 while guaranteeing no more than 3 incorrected bases, thus assuring an expected base error per read in each strand no larger than 2%. A feature table counting a total of 715,314 denoised sequences was obtained. SVs were assigned taxonomy using a machine learning algorithm based on the Naive-Bayes classifier model (‘qiime feature-classifier fit-classifier-naive-bayes’ Qiime2 plugin[86]) trained on the Silva138 99% database trimmed to the V4–V5 region of the 16S. The supervised trained classifier was then applied to the obtained SVs for mapping them to taxonomy by means of the ‘qiime feature-classifier classify-sklearn’ Qiime2 plugin. A phylogenetic tree was constructed via sequence alignment with MAFFT[87] (‘qiime alignment mafft’ Qiime2 plugin), filtering the alignment (‘qiime alignment mask’ Qiime2 plugin) and applying FastTree[88] (‘qiime phylogeny fasttree’ Qiime2 plugin) to generate the tree.

Denoised sequences were filtered according to the type of sample (feces or lungs). For the lungs, a further manual taxa filtering criterion has been applied according to Olesen et al.[89], thus removing taxonomies already classified as contaminants of the respiratory microbiome. i.e., ASVs classified no further than class-level, ASVs not classified as Bacteria (kingdom), and ASVs belonging to the phyla Cyanobacteria, Plantomycetes, Chloroflexi, and Deinococcus-Thermus, the class Rhodothermia, or the orders Rhizobiales, Rhodobacterales, Oceanospirillales, Azospirillales, and Rhodospirillales. Rare taxa, defined as sequence variants counted less than 10 times or present in no more than 3 samples, were then removed from feces and

lungs separately. After that, rarefaction strategy of samples was pursued in order to characterize the composition of each group of samples with respect to the same feature set and to guarantee that differences between groups are not due to different library size. The analysis of the rarefaction curves calculated for each sample with the ‘qiime diversity alpha-rarefaction’ plugin on 50 iterations of the Shannon and Chao1 indexes at SV level, indicates a good sequencing quality as both richness and number of rares and evenness does not increase significantly with the sampling depth (Fig. S4). Dashed black lines point to the sequencing depth of the sample with the fewest reads in each dataset, D*, the sequencing depth value that limits the proportion detectable in that sample. Boxplots of Shannon indexes on the 50 rarefactions evaluated just before D* show that each sample presents a very low variability around the median value thus supporting, for each iteration at depth D*, a stable composition within each sample. Also, boxplots of the Chao1 indexes rarefed just before D* present a relatively low variability for almost all samples, thus assessing that different rares does not affect significantly richness of each sample. Only a couple of samples present a wide Chao1 boxplot at depth D. In order to evaluate if rarefaction at D* affects composition variability of these few samples, also a beta rarefaction on 50 iterations of the Jaccard and of the Bray-Curtis dissimilarity matrixes has been evaluated by means of the ‘qiime diversity beta-rarefaction’ plugin and the corresponding first two PCoA components have been evaluated (Fig. S4). The PCoA plots show that uncertainty in microbe composition due to rarefaction iterations do not affect substantially the position, and thus the ordination, of samples neither in Jaccard or in Bray-Curtis indexes. In order to compare the 50 rarefied beta diversities outcomes, Mantel correlation (Spearman) test between rarefaction trials of each diversity matrix has been carried out. Correlation values for all trials is close to 1 for both metrics and for both datasets. It follows that both qualitative and quantitative composition of each sample is not significantly altered if a single rarefied iteration is considered. Thus, rarefaction has been used as normalization technique for evaluating differential analyses between groups in each dataset. Since high-throughput data are intrinsically compositional, the number of absolute reads obtained per sample is “irrelevant”[90] and thus feces samples were rarefied to 6700 reads, lung samples were rarefied to 3600 reads. This choice facilitates the comparisons between groups since all samples in a dataset refer to the same feature set. It should be noticed that sample rarefaction to 3600 reads, while allowing the largest number of samples to be included, could lead to an 8% worst-case relative error on the per-sample alpha index estimates[91]. Since groups of interest have at least three mice, the per-group average value of alpha indexes present a worst-case relative error reduced, with respect to the per-sample estimates, by a factor at least equal to sqrt(3), i.e., improved to <4.6%. Further analyses were conducted using both Qiime2 (ver. 2022.2) platform and R version 4.2.1 (2022-06-23) in RStudio 2022.02.3 + 492 “Prairie Trillium” Release (1db809b8323ba0a87c148d16eb84e-fe39a8e7785, 2022-05-20) for Ubuntu Bionic Mozilla/5.0 (Windows NT 10.0; Win64; x64; rv:108.0) Gecko/20100101 Firefox/108.0. Diversity indexes evaluation and differential analyses between groups were all carried out on rarefied feature tables at sequence variants. The within-sample alpha-diversity was assessed based on the 16S rRNA gene sequencing data, using Observed features, Chao1, and Shannon diversity indexes estimated by using the ‘qiime diversity alpha’ QIIME2 plugin applied to rarefied samples in each dataset. Statistical significances in sample groups comparison of the estimated alpha indexes were determined using a Kruskal-Wallis test calculated with the ‘qiime diversity alpha-group-significance’ tool. The beta-diversity was assessed on the SVs using Jaccard and Bray-Curtis distances estimated for all pairs of samples of each dataset in the rarefied table by the ‘qiime diversity beta’ tool with the FeatureTable[Frequency] semantic type as input. Principal coordinates matrices were computed from the resulting distance matrices by the ‘qiime diversity pcoa’ plugin

in order to separate quantitatively all sources contributing to the beta-diversity. To reduce the dimensionality of the diversity investigation, the principal coordinate analysis (PCoA) was limited to the first three components, thus allowing visualization of the most effective relationships contributing to diversity between groups of samples. Pairwise comparisons were assessed by non-parametric test (BH adjusted *p* value). The LEfSe (Linear discriminant analysis effect size)[92] was used on rarefied relative abundances tables to test the association between groups at each taxonomic level. LEfSe employs a non-parametric Kruskal-Wallis sum-rank test to differentiate between class features and a subsequent LDA to estimate effect size of taxa which violated the null hypothesis. LEfSe has been applied with default alpha values for the Anova and Wilcoxon test (0.1) and the LDA effect size has been evaluated by setting the absolute value of the logarithmic LDA threshold equal to 2.0. Other LEFSE parameters have been set to the default. Differential abundance between groups were evaluated also with ALDEx2 analyses using the Q2-ALDEx2-plugin with a Q-score significance threshold of 0.1[93]. Results were represented on volcano plots showing the opposite of the logarithm of the p values of the log-fold-change versus the log-fold-change itself. Labels on volcano's reveal bacterial sequence variants of those genera significantly associated by LEfSe to the corresponding group. The Phylogenetic Investigation of Communities by Reconstruction of Unobserved States (PICRUSt) pipeline was applied to predict metagenome functions from 16 S metagenomic samples. In particular, the enzyme-catalyzed reaction (EC number), functional gene content based on KEGG database annotations for reference genomes (KEGG Orthology), and metabolic pathway abundances of microbial communities using the pathway rules from MetCyc database were predicted with PICRUSt2[94]. Sequenced samples were provided as SV abundance. Bacterial sequence variants with an effect size associated by Q2-ALDEx2 to *Tph1*[−/−] at dpi 7 were used to create networks in Cytoscape 3.9.1[95] to gain insight into the interaction between bacterial sequence variants and selected enzymes (Supplementary Data file 1 and 2). To this purpose, Picrust2 was used to assess qualitatively which taxa contribute to each of the selected enzymes in the gut and lung, and the result was used to connect bacterial to enzymes nodes. Since Picrust2 outputs compositional data, rarefaction strategy of the EC predictions was applied, and centered log-ratio transform (CLR) was applied to rarefied predicted abundances for comparison purposes.

## Lung and fecal microbiota transplantation

For the lung microbiota transplantation, fresh lungs from *Tph1*[+/+] or *Tph1*[−/−] mice were collected, added with sterile PBS at 200 mg/mL, homogenized, and centrifuged at $400 \times g$ for 10 min at 4 °C. The supernatant was centrifuged at $13,400 \times g$ for 10 min at 4 °C and suspended pellet in PBS. For fecal microbiota transplantation, fresh fecal pellets from *Tph1*[+/+] or *Tph1*[−/−] mice were collected, added with sterile PBS at 10 mg/ml, homogenized, and centrifuged at $1500 \times g$ for 10 min at 4 °C. The lung (intranasal route) and fecal preparations (intragastrical route) were administrated once a day in concomitance with *A. fumigatus* infection. Mice were killed on day 4 and evaluated for infectious and metabolic parameters.

## CALUX cell analysis

To assess the activation of AhR, we used mouse hepatoma cells (H1L6.1c3), containing the stably integrated AhR xenobiotic responsive element driven by a firefly luciferase reporter plasmid, pGudLuc6.1[96]. Cells were plated and after 24 h at 37 °C, plated cells were stimulated with lung homogenates of infected or not mice for 24 h. After incubation cells were washed with PBS, 100 μL of 1× Lysis buffer (Tris phosphate 1 M pH 7.8, EDTA 0.5 M, glycerol 50%, Triton X and $H_2O$) was added to each well and plate placed on the plate shaker at room temperature until cells were lysed. Luciferase activity was measured using 100 μL of 1× Reaction Buffer (Reaction Buffer 2×, luciferin

10 mM, ATP 100 mM, CoA 10 mM, and $H_2O$) for 30 μL of cell lysate. Luciferase activity, normalized to sample protein concentration, was calculated as relative light units per microgram of protein and expressed as fold induction.

## Sample collection for LC-MS

Whole lungs were weighed and homogenized in 1.5 mL of PBS, were centrifuged at $12,000 \times g$ for 20 min, and stored at −80 °C until the analyses could be performed.

## Trp metabolites quantification by LC high-resolution mass spectrometry

Trp (Tryptophan), KYN (Kynurenire), KA (Kynurenic acid), XA (Xanthurenic acid), 5-HTP (5-Hydroxy-L-Tryptophan), 5-OH-IAA (5-Hydroxyindole-3-acetic acid), NAS (N-acetyl-5-Hydroxytryptamine), ILA (Indole-3-lactic acid), IPA (Indole-3-propionic acid), 3-IAld (Indole-3-carboxaldehyde), IS (Indoxyl sulfate), ICA (Indole-3-carboxylic acid) were obtained from Merck KGaA (Darmstadt, Germany). Labeled internal standards, $^{13}C_8$-3-IAld, 5-HT-d4, and 5-OH-IAA-d5 were from Toronto Research Chemicals (Toronto, Canada), Trp-d5 and ICA-d5 were purchase from Merck KGaA, KA-d5 was purchase from Santa Cruz (Dallas, Texas, USA). Individual stock solutions (100 mg mL⁻¹) were prepared in methanol. Two hundred microliters of lung homogenate were spiked with 10 μL of a solution at 1000 ng mL-1 of internal standard and extracted with 580 μL of acetonitrile. After vortexing and centrifugation (5 min, $16,000 \times g$), 600 μL of supernatant were transferred into a 2 mL tube and evaporated under nitrogen stream (40 °C). The sample was resuspended in 150 μL of a mixture $H_2O$/MeOH 95:5 (v/v), shaken, and then injected in a Liquid-Chromatography coupled to Quadrupole-Orbitrap mass spectrometer platform (LC-Q-Exactive Plus, Thermo Scientific, San Jose, CA, USA). Analyte separation was carried out using an Acquity BEH C18 (150 × 2.1 mm, 1.7 μm, Waters, Milford, MA, USA) connected with a C18 guard column (2.1 × 5 mm, Waters) and installed on an Ultimate 3000 UPLC system (Thermo Scientific). For the analysis of TRP, KYN, KA, XA, 3-IAld, 5-HTP, 5-OH-IAA, and NAS positive ionization mode (ESI+) was applied; the separation was achieved using the following mobile phases: water (A) and acetonitrile (B), both containing 0.1% formic acid. The flow rate was 0.3 mL min⁻¹. The gradient started with 0% B (2 min); then the percentage of B was increased to 30% at 10 min and to 40% at 11 min. This condition was maintained for 1 minute, then B was to 50% at 13 min and to 100% B at 13.5 min. This condition was retained for 2.5 min and the column was re-equilibrated to the initial condition for 3 min (total run time: 19 min). IPA, ILA, ICA, and IS were acquired in negative ionization mode (ESI−); the mobile phases were water (A) and methanol (B), both containing 5% ammonium acetate. The flow rate was 0.2 mL min⁻¹. The gradient started with 0% B and was maintained for 2 min; from 2 to 10 min, the % B increased to 100%. This condition was maintained for 2.5 min and then the system returned to 0% B (0.5 minute), and it was re-equilibrated for 2 min (total run time: 15 min). For both chromatographic runs the injection volume was 5 μL, the column and autosampler temperatures were kept at 40 °C and 16 °C, respectively. The mass spectrometer Q-Orbitrap was equipped with a heated electrospray ionization (HESI-II) source. The HESI-II and capillary temperatures were set at 350 °C and 300 °C, respectively, and the electrospray voltage at 3.50 kV (both in positive and negative ionization mode). Sheath and auxiliary gas flow rates were 40 and 15 arbitrary units, respectively. Data were acquired and processed with Xcalibur software version 3.0 (Thermo Scientific). The acquisition was achieved in full scan/dd-MS2. In negative ionization mode, the mass range was m/z 100–550, whereas in the positive one the range was m/z 110–1000. Resolving power was set at 35000 FWHM

(at m/z 200) in full scan mode. Automatic Gain Control (AGC) was set at $1 \times 10^6$ ions with a maximum injection time (IT) of 160 ms. A resolving power of 17500 FWHM (at m/z 200) was used for dd-MS2 experiments with an AGC target of $1 \times 10^6$ ions for a maximum IT of 80 ms. Stepped collision energy of 10, 30, and 50 (NCE) were applied in both ionization modes. Two inclusion lists were built to perform full-MS/dd-MS2 experiments in positive and negative ionization mode, respectively, reporting for each analyte the theoretical m/z of precursor ion and retention time (±1 min). The accuracy error threshold was fixed at 5 ppm.

### Statistical analysis

Unpaired $t$ test, Multiple $t$ test, one-way and two-way analysis of variance with Sidak or Turkey post hoc test was used to determine the statistical significance. Significance was defined as $p < 0.05$. Data are pooled results (mean ± SEM) or representative images from two or four (for the murine experiment) or two experiments (for the cells experiment). The histological scores were compared using the Kruskal-Wallis test with Dunn's multiple comparison test. The PCoA were compared using Kruskal–Wallis test and significance of the Wilcoxon multiple comparison. The in vivo groups consisted of three to six mice/group. Mice were allocated in each group by simple randomization. No criteria were set for including/excluding animals during the experiments. GraphPad Prism software V.6.01 (GraphPad Software) was used for analysis.

### Reporting summary

Further information on research design is available in the Nature Portfolio Reporting Summary linked to this article.

### Data availability

The datasets generated and/or analyzed during the current study have been deposited in the Sequence Read Archive (SRA) (https://www.ncbi.nlm.nih.gov/sra) under BioProject ID number PRJNA980163. The following databases were used: Silva138, KEGG, and MetaCyc. Source data are provided with this paper.

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

## Acknowledgements
Sources of support: supported by the European Union's Horizon 2020 research and innovation program under grant agreement no. 847507 to L.R.

## Author contributions
F.D.O. and G.R. designed the experiments, performed in vivo studies, and analyzed the data; M.P. helped with in vivo studies and performed histological analysis; R.G. and C.B. performed mass spectrometry analysis; C.A. and C.S. performed in vitro studies; M.M.B. performed western blot analysis; H.E. and C.L.-F. provided human CF samples; C.C., M.F., and V.N. helped with metagenomics; A.K.E. provided reagents and expertize; E.N. performed metagenomics analysis; E.G. and L.R. supervised the entire work and L.R., F.D.O. and G.R. wrote the manuscript.

## Competing interests
The authors declare no competing interests.
