## [Peer Review File · Nature Communications]

Bridging of host-microbiota tryptophan partitioning by the serotonin pathway in fungal pneumoniaREVIEWER COMMENTS

Reviewer #1 (Remarks to the Author):

The manuscript by D'Onofrio investigated tryptophan metabolism via the Trp hydroxylase/5-HT pathway during fungal pneumonia. Results showed that Tph1^{-/-} mice are more susceptible to lung infection with *Aspergillus fumigatus* via impaired neutrophil recruitment, defect macrophage phagocytosis, impaired Th1, Th2 and Treg responses and exacerbated Th17 responses. Similar results were observed with 5-HT depletion. 5-HT was produced by mast cells in the lung and drove IDO1 induction, whereas AhR activation was observed in Tph1^{-/-} mice. 5-HT deficiency was associated with alterations in both the lung and intestinal microbiota. 5-HT was found to be defective in CF mice and in humans with CF. The manuscript is well-designed and written and the results seem to support the conclusions.

Comments for the author's consideration.

1. Considering the level of impaired lung neutrophil recruitment (1b) and dissemination of *A. fumigatus* to the brain (1a), do the Tph1^{-/-} demonstrate increased mortality and if so, can this be modulated by 5-HT?
2. Are neutrophils from Tph1^{-/-} mice defective in ROS production, *A. fumigatus* killing or demonstrate impaired chemotaxis? Likewise, the Tph1^{-/-} mice have increased susceptibility to *A. fumigatus* in the presence of enhanced monocyte recruitment to the lung (1b). Monocytes have been shown to be potent effector cells against *A. fumigatus*, thus also provoking the question as to whether these cells are defective as well.
3. The T helper responses in Tph1^{-/-} are quite dramatic at 7 days post-infection (1i). However, the histological changes 3 days post-infection in WT Tph^{+/+} mice are also dramatic – what were the differences in T helper responses between the two strains at this time point?
4. Tph^{-/-} have aggravated AhR activation, yet it is not clear what elements of this activation are contributing to the susceptibility to *A. fumigatus*.
5. The 5-HT data in the human samples (9d) would likely benefit from normalization.
6. The Figure legends lack how many times the experiments were performed. For consistency purposes, it might be better if all graphs were columns or dot plots rather than intermingling the two formats.

Reviewer #2 (Remarks to the Author):

In their study D'Onofrio et al. investigated fungal burden, inflammatory response and Trp metabolism in mice with *A. fumigatus* induced pneumonia. The authors describe a role for mast cell derived serotonin in pathogen clearance and immune homeostasis in this setting. The authors attribute these effects to a role of the lung Tph1/5-HT pathway in coordinating the IDO1/AhR axis in *A. fumigatus* pneumonia via metabolite signaling and regulation of local and distal microbiota. The results of this study are very interesting and contribute to the understanding of the inflammatory effects of non-neuronal serotonin, which is of great importance in exploring its therapeutic potential.

Several points should be addressed by the authors:

1. The introduction should give the reader at least some background information on *A. fumigatus* pneumonia and its relevance.
2. Lines 42-44 of the introduction read: "Once released, 5-HT is transported into surrounding epithelial cells — or taken up by platelets in circulation — by the serotonin reuptake transporter (SERT) to be degraded to 5-hydroxyindoleacetic acid (5-OH-IAA)." This is misleading. Platelets store serotonin in their dense granules in high concentrations and release it upon platelet activation. Furthermore, please remove the word "reuptake" from the term "serotonin reuptake transporter" in line 43.
3. The entire manuscript should be checked for grammar and orthography. Some sentences are not understandable because words are missing, the sentences are too long or the correct context is missing (e.g., lines 91-93: what does "and increased expression of chemokine-recruiting

monocyte" mean?; lines 56-59: How is it prove for locally produced 5-HT if platelet- and not MC-derived serotonin is pivotal for allergic airway inflammation?; lines 154-158: One cannot follow the reasoning in this sentence because it is too long and complex.

4. Is there a word missing in the title of the manuscript? Bridging "of" (?) host-microbiota tryptophan partitioning by the serotonin pathway in fungal pneumonia

5. Fig. 1b: How exactly were the percentages of neutrophil and monocytes determined? Could a calculated percentage increase in monocytes only be relative to a decrease in neutrophils (or vice versa)? Why are there no absolute numbers given? Do absolute numbers differ significantly? Please clarify.

6. Fig. 2b: the fungal burden in both, Tph^{-/-} and Tph^{+/+} (without pretreatment with PCPA or 5-HTP), after *A. fumigatus* infection is much lower than shown in Fig. 1a. Why? Also, in Fig. 2b only data at 7 dpi are shown. However, the difference in fungal burden between Tph^{-/-} and Tph^{+/+} is much bigger at 3 dpi than at 7 dpi. How does PCPA or 5-HTP treatment of Tph^{+/+} and Tph^{-/-} mice, respectively, alter fungus burden at 3 dpi?

7. How does 5-HT (or precursor) treatment affect fungal burden and inflammation in Tph^{+/+} mice after infection? This would provide information on potential adverse effects.

8. Are Ido1 or AhR inhibitors/activators commercially available? To clearly show that differential Trp partitioning is the reason for the beneficial effect of serotonin, experiments involving such agents +/- serotonin or PCPA treatment are necessary.

9. Is a mouse model with MC specific Tph1 K.O. available? This would strengthen the assumption that locally produced MC-specific serotonin plays the decisive role in *A. fumigatus* infection.

10. Fig. 3B: Was monoaminoxidase expression altered over the course of disease and between genotypes?

11. Fig. 3C: Where plasma serotonin concentrations measured?

Reviewer #3 (Remarks to the Author):

In the manuscript entitled 'Bridging host-microbiota tryptophan partitioning by the serotonin pathway in fungal pneumonia', D'Onofrio et al. unraveled the protective role of Tph1/5-HT in controlling fungal growth and regulating lung inflammation and immunity by Trp utilization. Moreover, they linked the 16S rRNA gene sequencing profiles of mice to the coordination activity of 5-HT.

The study is of significance and novel, considering that the Role of Trp has mainly been studied in intestinal homeostasis so far, but I have concerns in regards to

(1) the analytical decision making and hence final interpretation of the data. Without more detailed information on the bioinformatics workflows and algorithms/packages/functions used, it is currently difficult to evaluate the statistical robustness. The detailed workflow, with version numbers, functions, packages and input arguments should be provided in the method section. However, I recommend to share all scripts in a public repository to enhance transparency and comparability across studies.

(2) the lack of negative controls in a study design focussing on lung samples with generally low microbial biomass.

(3) The low number of mice that went into parts of the analysis, including three mice for the network analysis.

Please see the following further comments:

Method section:

474: Six-to eight-week-old male and female mice weighing 20 to 25 g were 474 used in all experiments.

- How many mice? How many were male or female? Did you detect a sex bias?

580: DNA was isolated from murine lung and feces.

- 'Murine lung' refers to lung tissue? Which location or part of the lung was sampled?

- How was the lung tissue obtained? How was the tissue stored and at which temperature?

582: To extract bacterial DNA from lung tissue we performed both enzymatic and mechanical lysis

by the PowerBead tubes 1.4mm (Qiagen).

- Please provide more details. The experiments cannot be repeated based on the description. I strongly recommend publishing the detailed protocol in a public repository.

588: A total of 715,314 high quality sequences were obtained after trimming, paired-reads assembly and quality screening of raw reads obtained by Illumina MiSeq sequencing of the two bacterial 16S hypervariable regions V4 and V5. Only reads with at least 100 nucleotides (nt) were retained and then primer sequences were detected, clipped and oriented into forward-reverse primer orientation after removing the primer sequence using cutadapt. All sequenced files were subjected to a quality control procedure by using FASTQC software and then were imported and analyzed by using the next-generation microbiome bioinformatics platform Qiime2 platform (version 2022.2)

- How do you define 'high quality sequences'?

- How was the assembly and quality screening performed? What were quality thresholds and underlying filtering decisions?

- Was the assembly performed before clipping and removing the primer sequences and before the quality control procedure with FASTQC? Are you here providing a chronological ordering of the steps?

- Please share your scripts with all command line arguments, packages and versions in a public repository or give at least a more detailed overview on the workflow.

- To my understanding, the Qiime2 platform has dozens of different functions with hundreds of potential input parameters and arguments, which one did you chose?

610: In order to ease the comparison of microbial composition between groups, samples were normalized by rarefying the sequencing: feces samples were rarefied to 6700 reads, lung samples were rarefied to 3700 reads."

- So, recovering 3700 reads confirms that the lung tissue samples had generally low microbial biomass. How does your kitome look like? Which taxa did you detect in your negative controls? What is your splashome between samples? Which samples have been processed and sequenced together? [1, 2, 3]. Has this been done in a group-wise manner? Do you see a signal there? Are samples that were processed next to each other generally more similar? If no negative controls have been sampled and sequenced in parallel with the biological samples, this is a major limitation that requires discussion in the paper.

[1] Weyrich LS, Farrer AG, Eisenhofer R, Arriola LA, Young J, Selway CA, Handsley-Davis M, Adler CJ, Breen J, Cooper A. Laboratory contamination over time during low-biomass sample analysis. *Mol Ecol Resour.* 2019 Jul;19(4):982-996. doi: 10.1111/1755-0998.13011. Epub 2019 Apr 29. PMID: 30887686

[2] Salter SJ, Cox MJ, Turek EM, Calus ST, Cookson WO, Moffatt MF, Turner P, Parkhill J, Loman NJ, Walker AW. Reagent and laboratory contamination can critically impact sequence-based microbiome analyses. *BMC Biol.* 2014 Nov 12;12:87. doi: 10.1186/s12915-014-0087-z. PMID: 25387460

[3] Olomu IN, Pena-Cortes LC, Long RA, Vyas A, Krichevskiy O, Luellwitz R, Singh P, Mulks MH. Elimination of "kitome" and "splashome" contamination results in lack of detection of a unique placental microbiome. *BMC Microbiol.* 2020 Jun 11;20(1):157. doi: 10.1186/s12866-020-01839-y. PMID: 32527226

- Please also re-consider the rarefaction strategy. I know there is a large debate in the community whether to use rarefaction as normalization strategy in microbiome research or not, since valid data is omitted, artificial uncertainty is introduced by randomly sub-setting the samples and ecological diversity parameters get distorted [1,2,3]. If rarefaction remains the normalization strategy of choice here, I recommend performing at least repeated rarefaction which yields band of diversity values and hence accounts for different library sizes [4].

- Related to that, where rare taxa filtered out based on either abundance or prevalence? If so, was this filtering performed in a group-wise manner? All of these analytical decisions will affect the statistical comparison between time points, the alpha and beta diversity indices and are hence relevant information to compare the output of this study with the output of other available studies.

- Also, did you attenuate the compositionality effect of the 16S rRNA gene sequencing data and if so, how? [3]

[1] McMurdie PJ, Holmes S. Waste not, want not: why rarefying microbiome data is inadmissible. *PLoS Comput Biol.* 2014 Apr 3;10(4):e1003531. doi: 10.1371/journal.pcbi.1003531. PMID: 24699258; PMCID: PMC3974642.

[2] Lin H, Peddada SD. Analysis of microbial compositions: a review of normalization and differential abundance analysis. *NPJ Biofilms Microbiomes.* 2020 Dec 2;6(1):60. doi: 10.1038/s41522-020-00160-w. PMID: 33268781; PMCID: PMC7710733

[3] Gloor GB, Macklaim JM, Pawlowsky-Glahn V, Egozcue JJ. Microbiome Datasets Are Compositional: And This Is Not Optional. *Front Microbiol.* 2017 Nov 15;8:2224. doi: 10.3389/fmicb.2017.02224. PMID: 29187837; PMCID: PMC5695134.

[4] Cameron ES, Schmidt PJ, Tremblay BJ, Emelko MB, Müller KM. Enhancing diversity analysis by repeatedly rarefying next generation sequencing data describing microbial communities. *Sci Rep.* 2021 Nov 16;11(1):22302. doi: 10.1038/s41598-021-01636-1. PMID: 34785722; PMCID: PMC8595385.

LEFSE has been applied with default alpha values for the Anova and Wilcoxon test (0.05) and the LDA effect size has been evaluated by setting the absolute value of the logarithmic LDA threshold equal to 3.5. Other LEFSE parameters have been set to the default.

- Did you normalize the data beforehand with total sum scaling or which normalization, if any, has been used prior to running LEfSe?
- Why was a threshold of 3.5 chosen?
- What were the groups under comparison?

Result section:

Figure 1-4:

- Instead of bar plots, I would recommend showing the real data points. Particularly for experiments with less than 5 mice. The statistical comparison between those groups has generally low power. This should be clearly visible and not hidden in the bar representation.

Figure 5 and Figure 7.

- Is the PCoA diversity analysis on Phylum or Genus or Species level? Is the Jaccard distance based on a presense/absence matrix? Was the presence/absence matrix established from rarified count data?
- What is the rationale for performing the LEfSe at genus level? With 16S rRNA V3-V4 it should be possible to get a slightly higher taxonomic resolution. I am a bit confused. In the figure legend, it is stated the LEfSe has been done on genus level, but the taxonomic description and color code is a mixture of taxonomic ranks including order (e.g. Pseudomonadales) and family (e.g., Peptostreptococcaceae) ranks.
- How large is the within-group variability you observed? The standard error ellipses are large and mainly overlapping. There seems to be more variance in one group than between groups. Therefore, the interpretation is not yet completely convincing.

Figure 6.

- How many mice per group went into the analysis? Was the data normally distributed enabling you to perform a two-way ANOVA analysis? Or what was the rationale to choose ANOVA testing from a statistical point of view?

Figure 8.

- How was the network analysis performed? I do not see any information in the method section. How is presence/absence defined? How confident are you in the networks based on the number of mice that were available for the analysis and based on the rarefaction normalization strategy that was applied before? What if pathways or taxa were low abundant in some samples and have been cut off by the random subsampling? What is the advantage of performing the analysis on family level?

Point-by-point reply letter

Reviewer #1 (Remarks to the Author):

The manuscript by D'Onofrio investigated tryptophan metabolism via the Trp hydroxylase/5-HT pathway during fungal pneumonia. Results showed that Tph1^{-/-} mice are more susceptible to lung infection with *Aspergillus fumigatus* via impaired neutrophil recruitment, defect macrophage phagocytosis, impaired Th1, Th2 and Treg responses and exacerbated Th17 responses. Similar results were observed with 5-HT depletion. 5-HT was produced by mast cells in the lung and drove IDO1 induction, whereas AhR activation was observed in Tph1^{-/-} mice. 5-HT deficiency was associated with alterations in both the lung and intestinal microbiota. 5-HT was found to be defective in CF mice and in humans with CF. The manuscript is well-designed and written and the results seem to support the conclusions.

Comments for the author's consideration.

Comment n. 1. Considering the level of impaired lung neutrophil recruitment (1b) and dissemination of *A. fumigatus* to the brain (1a), do the Tph1^{-/-} demonstrate increased mortality and if so, can this be modulated by 5-HT?

Response n. 1. The new Figure 2 incorporates (panel b) the data on mortality obtained by injecting a higher inoculum of *Aspergillus* conidia. The data clearly showed how the increased mortality observed in Tph1^{-/-} mice could be reduced by treatment with 5-HTP.

Comment n. 2. Are neutrophils from Tph1^{-/-} mice defective in ROS production, *A. fumigatus* killing or demonstrate impaired chemotaxis? Likewise, the Tph1^{-/-} mice have increased susceptibility to *A. fumigatus* in the presence of enhanced monocyte recruitment to the lung (1b). Monocytes have been shown to be potent effector cells against *A. fumigatus*, thus also provoking the question as to whether these cells are defective as well.

Response n. 2. Similar to what observed with recruited monocytes, phagocytosis and not the conidiocidal or the oxidative activities were impaired in neutrophils from Tph1^{-/-} mice (see the new panel e of Fig. 1 and a mention in the text, page 4 line 98, Figure 1e).

Comment n. 3. The T helper responses in Tph1^{-/-} are quite dramatic at 7 days post-infection (1i). However, the histological changes 3 days post-infection in WT Tph^{+/+} mice are also dramatic – what were the differences in T helper responses between the two strains at this time point?

Response n. 3. As per our experience, the expression of the Th transcription factors is not informative at this time point of the infection. Therefore, we have measured the cytokine levels in the lung homogenates at this time point and the results have been incorporated in the new Figure 1 (panel i) and mentioned in the text (page 4, lines 101-105, Figure 1i).

Comment n. 4. Tph^{-/-} have aggravated AhR activation, yet it is not clear what elements of this activation are contributing to the susceptibility to *A. fumigatus*.

Response n. 4. Like this reviewer, we were much intrigued about this finding for a better understanding of which we have resorted to a custom QuantiGene Plex gene expression assay (Thermo Fisher Scientific) in order to assess AhR-related genes in *Tph*^{-/-} mice. Among the genes differently regulated in *Tph*^{+/+} and *Tph*^{-/-} littermates, we found that the *Serpinb2* gene was upregulated in *Tph*^{-/-} but not in *Tph*^{+/+} or *AhR*^{-/-} mice (Figure below).

Figure legend. The figure shows the normalized mRNA expression values of of the selected genes on total lung cells from the indicated mouse strains at different days (dpi) after the *A. fumigatus* infection as described in Material and Methods.

Because the *Serpinb2* gene expression by AhR occurs through alternative mechanisms not involving the AhR nuclear translocator (Sekine et al., *Hypersensitivity of aryl hydrocarbon receptor-deficient mice to lipopolysaccharide-induced septic shock. Mol Cell Biol* 29:6391, 2019; Brauze et al. *Expression of Serpin Peptidase Inhibitor B2 (SERPINB2) is regulated by Aryl hydrocarbon receptor (AhR). Chem Biol Interact* 309:108700, 2019), we have taken this observation to indicate that a distinct AhR signaling pathway may occur in *Tph1*^{-/-} mice eventually leading and/or associated to an inflammatory phenotype. As a matter of fact, SerpinB2 is substantially up-regulated under multiple inflammatory conditions, and dysregulated expression and polymorphisms are associated with several human inflammatory diseases (Wayne et al. *The role of SerpinB2 in immunity. Crit Rev Immunol* 31:15-30, 2011) including senescence (Liao et al., *Unveiling a novel serpinB2-tripeptidyl peptidase II signaling axis during senescence. J Cell Sci* 135:jcs259513, 2022). In the lung, a genome-wide profiling study has identified SerpinB2 among the epithelial signatures of type 2 airways (Woodruff et al., *Genome-wide profiling identifies epithelial cell genes associated with asthma and with treatment response to corticosteroids. Proc Natl Acad Sci U S A* 104:15858, 2007). Of interest, a recent study by the Prof. Alberto Mantovani's group (Nina Cortese, Roberta Carriero, Marialuisa Barbagallo, Anna Rita Putignano, Guido Costa, Fabio Giavazzi, Fabio Grizzi, Fabio Pasqualini, Clelia Peano, Gianluca Basso, Sergio Marchini, Federico Simone Colombo, Cristiana Soldani, Barbara Franceschini, Luca Di Tommaso, Luigi Terracciano, Matteo Donadon, Guido Torzilli, Paolo Kunderfranco, Alberto Mantovani, Federica Marchesi. "High-resolution analysis of mononuclear phagocytes reveals GPNMB as a prognostic marker in human colorectal liver metastasis" *Cancer Immunology Research*, in press), identified a population of pro-inflammatory, less mature macrophages expressing the monocytic marker SERPINB2 along with the expression of genes related to leukocyte and neutrophil recruitment and cytokine production. At this point, mice homozygous for a targeted mutation of *Serpin2b*, known to have a slight to mild reduction in platelet and white blood cell number, will be crucial to unravel the role of SerpinB2 in lung pathology. An additional interesting finding that we have made is the lower expression of the pregnane-x-receptor (PXR)-dependent genes in *Tph*^{-/-} than *Tph*^{+/+} mice. Because AhR activation in *PXR*^{-/-} mice promotes lung damage and inflammation (our own unpublished observation), we have postulated that the role of PXR and AhR may go beyond the xenobiotics metabolism to include a fine regulation of the inflammatory response in the lung via a mutual cross-talk that is likely dysregulated in *Tph*^{-/-} mice (also mentioned in the discussion). Preliminary as they are, these observations are currently being investigated in our laboratory. Ultimately, in the relative absence of strong experimental evidence, we have clearly stated in the discussion that whether the defective IDO1 activation or the exuberant AhR activity or both contribute to the inflammatory tone in *Tph1*^{-/-} mice has to be investigated. Should this reviewer believe that these preliminary data are to be shown in a Supplementary figure, we will be happy to do so.

Comment n. 5. The 5-HT data in the human samples (9d) would likely benefit from normalization.

Response n. 5. Done.

Comment n. 6. The Figure legends lack how many times the experiments were performed. For consistency purposes, it might be better if all graphs were columns or dot plots rather than intermingling the two formats.

Response n. 6. The Figure legends have been updated and the graphs adjusted. We would like to clarify that the bar plots of panels e and i of Figure 1 were deliberately not modified for the loss of clarity.

Reviewer #2 (Remarks to the Author):

In their study D'Onofrio et al. investigated fungal burden, inflammatory response and Trp metabolism in mice with *A. fumigatus* induced pneumonia. The authors describe a role for mast cell derived serotonin in pathogen clearance and immune homeostasis in this setting. The authors attribute these effects to a role of the lung *Tph1*/5-HT pathway in coordinating the IDO1/AhR axis in *A. fumigatus* pneumonia via metabolite signaling and regulation of local and distal microbiota. The results of this study are very interesting and

contribute to the understanding of the inflammatory effects of non-neuronal serotonin, which is of great importance in exploring its therapeutic potential.

Several points should be addressed by the authors:

Comment n. 1. The introduction should give the reader at least some background information on *A. fumigatus* pneumonia and its relevance.

Response n. 1. Done (pag. 3, lines 75-76), along with a new reference (n. 23).

Comment n. 2. Lines 42-44 of the introduction read: "Once released, 5-HT is transported into surrounding epithelial cells — or taken up by platelets in circulation — by the serotonin reuptake transporter (SERT) to be degraded to 5-hydroxyindoleacetic acid (5-OH-IAA)." This is misleading. Platelets store serotonin in their dense granules in high concentrations and release it upon platelet activation. Furthermore, please remove the word "reuptake" from the term "serotonin reuptake transporter" in line 43.

Response n. 2. We have modified the sentence as follows: "Once released, 5-HT is transported into surrounding epithelial cells — or taken up by platelets in circulation — by the serotonin transporter (SERT). The word "reuptake" has been removed.

Comment n. 3. The entire manuscript should be checked for grammar and orthography. Some sentences are not understandable because words are missing, the sentences are too long or the correct context is missing (e.g., lines 91-93: what does "and increased expression of chemokine-recruiting monocyte" mean?; lines 56-59: How is it prove for locally produced 5-HT if platelet—and not MC—derived serotonin is pivotal for allergic airway inflammation?; lines 154-158: One cannot follow the reasoning in this sentence because it is too long and complex.

Response n. 3. Page 4, line 94, the sentence has been corrected. Page 5, lines 1242-145, the sentence has been amended to make it clear the concept of a different and distinct role MC-derived 5-HT may have in the lung immune homeostasis.

Comment n. 4. Is there a word missing in the title of the manuscript? Bridging "of" (?) host-microbiota tryptophan partitioning by the serotonin pathway in fungal pneumonia

Response n. 4. We have include "of" as suggested. Thank you.

Comment n. 5. Fig. 1b: How exactly were the percentages of neutrophil and monocytes determined? Could a calculated percentage increase in monocytes only be relative to a decrease in neutrophils (or vice versa)? Why are there no absolute numbers given? Do absolute numbers differ significantly? Please clarify.

Response n. 5. The absolute numbers are now given (Figure 1, panel b).

Comment n. 6. Fig. 2b: the fungal burden in both, Tph^{-/-} and Tph^{+/+} (without pretreatment with PCPA or 5-HTP), after *A. fumigatus* infection is much lower than shown in Fig. 1a. Why? Also, in Fig. 2b only data at 7 dpi are shown. However, the difference in fungal burden between Tph^{-/-} and Tph^{+/+} is much bigger at 3 dpi than at 7 dpi. How does PCPA or 5-HTP treatment of Tph^{+/+} and Tph^{-/-} mice, respectively, alter fungus burden at 3 dpi?

Response n.6. The new Figure 2c now provides novel data obtained with PCPA and 5-HTP at both day 3 and 7 dpi.

Comment n. 7. How does 5-HT (or precursor) treatment affect fungal burden and inflammation in Tph^{+/+} mice after infection? This would provide information on potential adverse effects.

Response n. 7. The figure below clearly shows that 5-HTP adversely affect the course of the infection upon administration to *Tph^{+/+}* mice. Because mention to 5-HT administration in healthy subjects has not been discussed in the present study, these data have not been included in the revised manuscript, unless the reviewer suggests to do so. In that case, we will be happy to follow.

Figure legend. The figure shows (a) the increased fungal growth and (b) lung inflammatory pathology (magnified in the inset) in *Tph1^{+/+}* mice treated with 5-HTP (see legend to Figure 2 for experimental details). Bars indicate 10x and 40x magnifications.

Comment n. 8. Are Ido1 or AhR inhibitors/activators commercially available? To clearly show that differential Trp partitioning is the reason for the beneficial effect of serotonin, experiments involving such agents +/- serotonin or PCPA treatment are necessary.

Response n. 8. We have done the experiments as suggested and the data are now provided in Figure S3 and mentioned in the text (Page 6, lines 166-168; page 14, lines 438-441 and Figure S3). We would like to add here that the results we have obtained are consistent with the results we are having on a continuation project in which mice with combined deficiency of each Trp pathway, alone or in combination, have been generated and assessed for the susceptibility to fungal pneumonia.

Comment n. 9. Is a mouse model with MC specific *Tph1* K.O. available? This would strengthen the assumption that locally produced MC-specific serotonin plays the decisive role in *A. fumigatus* infection.

Response n. 9. Agree with the reviewer. Unfortunately, to our knowledge, these mice are not available.

Comment n. 10. Fig. 3B: Was monoaminoxidase expression altered over the course of disease and between genotypes?

Response n. 10. Yes. The expression of monoaminoxidase A is now given in Figures 3 and 10 and mentioned in the text.

Comment n. 11. Fig. 3C: Where plasma serotonin concentrations measured?

Response n. 11. Yes, we measured it in the platelet-rich plasma collected with EDTA and it was not increased either.

Reviewer #3 (Remarks to the Author):

In the manuscript entitled 'Bridging host-microbiota tryptophan partitioning by the serotonin pathway in fungal pneumonia', D'Onofrio et al. unraveled the protective role of *Tph1*/5-HT in controlling fungal growth and regulating lung inflammation and immunity by Trp utilization. Moreover, they linked the 16S rRNA gene sequencing profiles of mice to the coordination activity of 5-HT.

The study is of significance and novel, considering that the Role of Trp has mainly been studied in intestinal homeostasis so far, but I have concerns in regards to

Comment n. 1. The analytical decision making and hence final interpretation of the data. Without more detailed information on the bioinformatics workflows and algorithms/packages/functions used, it is currently difficult to evaluate the statistical robustness. The detailed workflow, with version numbers, functions, packages and input arguments should be provided in the method section. However, I recommend to share all scripts in a public repository to enhance transparency and comparability across studies.

Response n. 1. We have taken into the proper consideration all the above suggestions and modified text and figures, accordingly. The raw data have been uploaded into the NCBI SRA. BioProject ID PRJNA980163 (<https://dataview.ncbi.nlm.nih.gov/object/PRJNA980163?reviewer=q0klo70j8us29acgfd3nqb6ar>) database while the section on microbiome analysis has been completely revised to provide a more detailed overview on the workflow.

Comment n. 2. The lack of negative controls in a study design focussing on lung samples with generally low microbial biomass.

Response n. 2. The figure below shows the negative results obtained upon quantification and amplification of negative controls as compared to a representative lung sample.

Comment n. 3. The low number of mice that went into parts of the analysis, including three mice for the network analysis.

Response n.3. As clearly stated in the MM, the in vivo groups consisted of three to six mice/group. The metagenomics analysis, including the network analysis, was done on 4 to 6 samples.

Please see the following further comments:

Method section:

Line 410: Six-to eight-week-old male and female mice weighing 20 to 25 g were used in all experiments.

Comment n. 4. How many mice? How many were male or female? Did you detect a sex bias?

Response n. 4. As mentioned above, the in vivo groups consisted of three to six mice/group (specified in the MM and the figure legends). Although we are much aware that gender may influence the incidence of some fungal infections, we did not have, so far, evidence of sex bias in our long-lasting experience with experimental fungal infection models, although we do use female preferably for the lower aggressive behavior. However, being the number of newborn +/+ and -/- mice always small and never enough, we have used either one interchangeably.

Line 580: DNA was isolated from murine lung and feces.

Comment n. 5. Murine lung' refers to lung tissue? Which location or part of the lung was sampled? How was the lung tissue obtained? How was the tissue stored and at which temperature?

Response n. 5. The entire lung was placed in ethanol 70% and stored at -80°C (page 17, line 523).

Line 582: To extract bacterial DNA from lung tissue we performed both enzymatic and mechanical lysis by the PowerBead tubes 1.4mm (Qiagen).

Comment n. 6. Please provide more details. The experiments cannot be repeated based on the description. I strongly recommend publishing the detailed protocol in a public repository.

Response n. 6. We revised the description of bacterial DNA extraction to include more details on the procedure (Page 17, line 521-527). Specifically: "DNA was isolated from murine feces by the QIAamp PowerFecal Pro DNA kit (Qiagen) according to the manufacturer's instructions. In the case of lung tissues, the following modification was introduced. Specifically, 20 mg of tissue were added to PowerBead tubes 1.4mm (Qiagen) and resuspended in 800 ul of CD1 solution. Mechanical lysis was performed with a bead-beater (BIO101/Savant FastPrep FP120) at 4000 rpm for 20 sec, followed by 20 sec in ice. The mechanical lysis was repeated twice before centrifugation and transfer to the PowerBead Pro Tubes." Since the extraction was performed by following the manufacturer's instructions with very minor modifications, publishing the detailed protocol in a public repository may perhaps not be necessary.

Line 588: A total of 715,314 high quality sequences were obtained after trimming, paired-reads assembly and quality screening of raw reads obtained by Illumina MiSeq sequencing of the two bacterial 16S hypervariable regions V4 and V5. Only reads with at least 100 nucleotides (nt) were retained and then primer sequences were detected, clipped and oriented into forward-reverse primer orientation after removing the primer sequence using cutadapt. All sequenced files were subjected to a quality control procedure by using FASTQC software and then were imported and analyzed by using the next-generation microbiome bioinformatics platform Qiime2 platform (version 2022.2)

Comment n. 7. How do you define 'high quality sequences?'

Response n. 7. The sentence (pages 19 and 20) has been completely revised and quality filtering criteria of reads in each strand have now been explicitly indicated.

Comment n. 8. How was the assembly and quality screening performed? What were quality thresholds and underlying filtering decisions?

Response n. 8. The sentence (pages 19 and 20) has been completely revised since the assembly procedure has not been applied. Moreover, the quality filtering criteria have now been explicitly indicated.

Comment n. 9. Was the assembly performed before clipping and removing the primer sequences and before the quality control procedure with FASTQC? Are you here providing a chronological ordering of the steps?

Response n. 9. The assembly procedure has not been applied. The description is now respecting the chronological ordering of steps (pages 19 and 20).

Comment n. 10. Please share your scripts with all command line arguments, packages and versions in a public repository or give at least a more detailed overview on the workflow.

Response n. 10. The raw data have been uploaded into the NCBI SRA. BioProject ID PRJNA980163 (<https://dataview.ncbi.nlm.nih.gov/object/PRJNA980163?reviewer=q0klo70j8us29acgfd3nqb6ar>) database while the section on microbiome analysis has been completely revised to provide a more detailed overview on the workflow.

Comment n. 11. To my understanding, the Qiime2 platform has dozens of different functions with hundreds of potential input parameters and arguments, which one did you chose?

Response n. 11. The Qiime2 plugins used for evaluating all bioinformatic steps now have been detailed and description of parameters other than the default have been made explicit (pages 19 and 20).

Line 610: In order to ease the comparison of microbial composition between groups, samples were normalized by rarefying the sequencing: feces samples were rarefied to 6700 reads, lung samples were rarefied to 3700 reads."

Comment n. 12. So, recovering 3700 reads confirms that the lung tissue samples had generally low microbial biomass. How does your kitome look like? Which taxa did you detect in your negative controls? What is your splashome between samples? Which samples have been processed and sequenced together? [1, 2, 3]. Has this been done in a group-wise manner? Do you see a signal there? Are samples that were processed next to each other generally more similar? If no negative controls have been sampled and sequenced in parallel with the biological samples, this is a major limitation that requires discussion in the paper.

[1] Weyrich LS, Farrer AG, Eisenhofer R, Arriola LA, Young J, Selway CA, Handsley-Davis M, Adler CJ, Breen J, Cooper A. Laboratory contamination over time during low-biomass sample analysis. *Mol Ecol Resour.* 2019 Jul;19(4):982-996. doi: 10.1111/1755-0998.13011. Epub 2019 Apr 29. PMID: 30887686

[2] Salter SJ, Cox MJ, Turek EM, Calus ST, Cookson WO, Moffatt MF, Turner P, Parkhill J, Loman NJ, Walker AW. Reagent and laboratory contamination can critically impact sequence-based microbiome analyses. *BMC Biol.* 2014 Nov 12;12:87. doi: 10.1186/s12915-014-0087-z. PMID: 25387460

[3] Olomu IN, Pena-Cortes LC, Long RA, Vyas A, Krichevskiy O, Luellwitz R, Singh P, Mulks MH. Elimination of "kitome" and "splashome" contamination results in lack of detection of a unique placental microbiome. *BMC Microbiol.* 2020 Jun 11;20(1):157. doi: 10.1186/s12866-020-01839-y. PMID: 32527226

Response n. 12. Contaminating bacterial DNA in the extraction kits have not been determined because, as previously mentioned, they were below the quantification limit and did not amplify in PCR. Samples were sequenced in a group-wise manner, with lung samples sequenced sequentially followed by fecal samples. Samples are more similar within rather than across experimental groups, indicating that a sample is more similar to the samples of the same experimental group rather than to the adjacent ones belonging to a different group. For the negative controls, see the previous reply.

Comment n. 13. Please also re-consider the rarefaction strategy. I know there is a large debate in the community whether to use rarefaction as normalization strategy in microbiome research or not, since valid data is omitted, artificial uncertainty is introduced by randomly sub-setting the samples and ecological diversity parameters get distorted [1,2,3]. If rarefaction remains the normalization strategy of choice here, I

recommend performing at least repeated rarefaction which yields band of diversity values and hence accounts for different library sizes [4].

Response n. 13. We thank the reviewer for this comment that gives us the opportunity to explain the choice of the rarefaction as normalization strategy for the scope of the paper. The paper has been integrated with alpha (Shannon and Chao1) rarefaction curves and PCoA of beta (Bray-Curtis and Jaccard) diversities rarefied for 50 times (Figure S4). This analysis shows that rarefaction bands of diversities values evaluated at sequencing depth equal to D^* (i.e. 3600 for lungs and 6700 for feces) remain separated for each sample at sequence variant level for both feces and lungs. Thus, the compositional analysis carried out on a single rarefied iteration is not affected significantly by the different library size of samples and does not change the ordination of samples. To corroborate this outcome, a Mantel correlation test between rarefaction trials of diversity matrixes for comparing the 50 rarefied diversities outcomes was also executed and it confirms that estimated Spearman correlation between rarefaction trials is almost equal to 1.

Comment n. 14. Related to that, where rare taxa filtered out based on either abundance or prevalence? If so, was this filtering performed in a group-wise manner? All of these analytical decisions will affect the statistical comparison between time points, the alpha and beta diversity indices and are hence relevant information to compare the output of this study with the output of other available studies.

Response n. 14. We are grateful to the reviewer for the observation that allow us to increase the comparability of our outcomes with other studies. Comments on rare taxa processing have been added in the text (pages 19 and 20) as follows: "Rare taxa, defined as sequence variants counted less than 10 times or present in no more than 3 samples, were then removed from feces and lungs separately". For the lungs, a further manual taxa filtering criterion has been applied according to Olesen *et al.* (Olesen *et al.*, *Changes in Skin and Nasal Microbiome and Staphylococcal Species Following Treatment of Atopic Dermatitis with Dupilumab. Microorganisms. 2021 Jul 13;9(7):1487*). Moreover, the following sentence has been inserted "Diversity indexes evaluation and differential analyses were all carried out on rarefied feature tables" (pages 19 and 20).

Comment n. 15. Also, did you attenuate the compositionality effect of the 16S rRNA gene sequencing data and if so, how? [3]

[1] McMurdie PJ, Holmes S. Waste not, want not: why rarefying microbiome data is inadmissible. *PLoS Comput Biol.* 2014 Apr 3;10(4):e1003531. doi: 10.1371/journal.pcbi.1003531. PMID: 24699258; PMCID: PMC3974642.

[2] Lin H, Peddada SD. Analysis of microbial compositions: a review of normalization and differential abundance analysis. *NPJ Biofilms Microbiomes.* 2020 Dec 2;6(1):60. doi: 10.1038/s41522-020-00160-w. PMID: 33268781; PMCID: PMC7710733

[3] Gloor GB, Macklaim JM, Pawlowsky-Glahn V, Egozcue JJ. Microbiome Datasets Are Compositional: And This Is Not Optional. *Front Microbiol.* 2017 Nov 15;8:2224. doi: 10.3389/fmicb.2017.02224. PMID: 29187837; PMCID: PMC5695134.

[4] Cameron ES, Schmidt PJ, Tremblay BJ, Emelko MB, Müller KM. Enhancing diversity analysis by repeatedly rarefying next generation sequencing data describing microbial communities. *Sci Rep.* 2021 Nov 16;11(1):22302. doi: 10.1038/s41598-021-01636-1. PMID: 34785722; PMCID: PMC8595385.

Response n. 15. The intrinsic compositional nature of the Illumina System has been considered by using data rarefaction (Figure S4) and adding the Q2-ALDEx2 analysis based on CLR for group comparison (Figures 6 and 8).

Comment n. 16. LEfSE has been applied with default alpha values for the Anova and Wilcoxon test (0.05) and the LDA effect size has been evaluated by setting the absolute value of the logarithmic LDA threshold equal to 3.5. Other LEfSE parameters have been set to the default. Did you normalize the data beforehand with total sum scaling or which normalization, if any, has been used prior to running LEfSe? Why was a threshold of 3.5 chosen? What were the groups under comparison?

Response n. 16. LEfSe input data were rarefied on absolute abundance table as described above and further normalized to relative abundances with total sum scaling. We inserted the following sentence: "The LEfSe (Linear discriminant analysis effect size) was used on rarefied relative abundances tables to test the association between groups at each taxonomic level". LDA threshold was reset to the default value (2.0) for all comparisons and alpha values of the Anova and Wilcoxon test were set to 0.1 in order to increase the confidence band. LEfSe outcome have been updated accordingly in the paper (pages 19-20).

Comment n. 17. Figure 1-4: Instead of bar plots, I would recommend showing the real data points. Particularly for experiments with less than 5 mice. The statistical comparison between those groups has generally low power. This should be clearly visible and not hidden in the bar representation.

Response n. 17. Modified as suggested by this reviewer and reviewer n. 1. We would like to clarify that the bar plots of panels e and i of Figure 1 were deliberately not modified for the loss of clarity.

Comment n. 18. Figure 5 and Figure 7. Is the PCoA diversity analysis on Phylum or Genus or Species level? Is the Jaccard distance based on a presense/absence matrix? Was the presence/absence matrix established from rarified count data?

Response n. 18. All diversity indexes have been evaluated on tables rarefied per organ, i.e. we rarefied all feces samples and all lung samples separately and evaluated diversity indexes for all samples in each of the two rarefied tables. It follows that, within each organ, we can compare diversity indexes since they evaluate the compositional diversity with respect to the same microbiome. The PCoA diversity analysis was done on sequence variants that improved diversity resolution.

Comment n. 19. What is the rationale for performing the LEfSe at genus level? Because no further valuable information were derived by a different analysis. With 16S rRNA V3-V4 it should be possible to get a slightly higher taxonomic resolution. I am a bit confused. In the figure legend, it is stated the LEfSe has been done on genus level, but the taxonomic description and color code is a mixture of taxonomic ranks including order (e.g. Pseudomonadales) and family (e.g., Peptostreptococaceae) ranks.

Response n. 19. We have modified the association analysis in order to improve consistency and integration of the data from genus analysis by LEfSE to sequence variant analysis by ALDEx2 (see the new Figures 6 and 8 and Tables S5 and S6).

Comment n. 20. How large is the within-group variability you observed? The standard error ellipses are large and mainly overlapping. There seems to be more variance in one group than between groups. Therefore, the interpretation is not yet completely convincing.

Response n. 20. The standard error ellipses have been eliminated to better emphasize the differences shown in the box-plots and Table S3.

Comment n. 21. Figure 6. How many mice per group went into the analysis? Was the data normally distributed enabling you to perform a two-way ANOVA analysis? Or what was the rationale to choose ANOVA testing from a statistical point of view?

Response n. 21. As indicated in the figure legend, we have used 3 control mice and between 7 to 12 mice in the treated group. The data were analyzed with the non parametric Dunn test and not with the two-way ANOVA as erroneously written in the figure legend. We apologize for this oversight.

Comment n. 22. Figure 8 - How was the network analysis performed? I do not see any information in the method section. How is presence/absence defined? How confident are you in the networks based on the number of mice that were available for the analysis and based on the rarefaction normalization strategy that was applied before? What if pathways or taxa were low abundant in some samples and have been cut off by the random subsampling? What is the advantage of performing the analysis on family level?

Response n 22. Network representation was performed with Cytoscape 3.9.1 and corresponding information has been added in the method section. Figure 8 (now Figure 10) has been improved by grouping bacterial sequence variants at genus level coherently with the LEFSE analyses, and by emphasizing that edges of the network indicate the qualitative contribution of the taxon to the selected enzymes. Concerning low abundance taxa or pathways, please note that being 16S data compositional, so are the Picrust2 predictions, and thus rarefaction of predicted data and corresponding CLR have been used in comparison procedures (see comments on [https://github.com/picrust/picrust2/wiki/PICRUST2-Tutorial-\(v2.5.2\)](https://github.com/picrust/picrust2/wiki/PICRUST2-Tutorial-(v2.5.2))).

We thank this reviewer for giving us the opportunity to improve the methodological part of the bioinformatic section, the workflow description and eliminate the typos.

REVIEWERS' COMMENTS

Reviewer #1 (Remarks to the Author):

The authors were responsive to the previous comments and have strengthened the manuscript.

Reviewer #2 (Remarks to the Author):

Thank you for this thorough revision.

All of my issues have been resolved and I have no further comments.

Reviewer #3 (Remarks to the Author):

I thank the authors for the comprehensive update of the method section and for offering more in-depth insights into the bioinformatics workflow. Although many of my queries have already been adequately resolved, I still have the following concerns:

Response n. 2. The figure below shows the negative results obtained upon quantification and amplification of negative controls as compared to a representative lung sample.

- I am pleased to note that the negative controls display very low biomass. Yet, I see only one lung sample selected as representative for all the lung samples. Were the quantities consistent across lung samples, or were there instances of lung samples with lower biomass? Either way, it remains pertinent to sequence blank controls, with or without artificial spike-ins, if required. If this step was skipped, I strongly suggest acknowledging it as a major limitation in your discussion.
- Kindly refer to Robert Dickson's publication that underscores the complexities associated with lung tissue and sequenced blank controls: Baker, J.M., Hinkle, K.J., McDonald, R.A., et al. "Whole lung tissue is the preferred sampling method for amplicon-based characterization of murine lung microbiota." *Microbiome* 9, 99 (2021). [<https://doi.org/10.1186/s40168-021-01055-4>]

Response n. 11. The Qiime2 plugins used for evaluating all bioinformatic steps now have been detailed and description of parameters other than the default have been made explicit (pages 19 and 20). Line 610: In order to ease the comparison of microbial composition between groups, samples were normalized by rarefying the sequencing: feces samples were rarefied to 6700 reads, lung samples were rarefied to 3700 reads."

- Please elaborate in the manuscript on the low number of reads available after rarefaction, especially in the light of the study by Burkin et al. (2019). Their results suggest that to achieve statistically convergent estimates of species diversity for most of the community, one should consider samples ranging from 10,000 to 15,000 reads. In such scenarios, the relative error for the diversity index estimates was less than 4%. How does this align with your outcome? See: Bukin, Y. S. et al. The effect of 16S rRNA region choice on bacterial community metabarcoding results. *Sci. Data*. 6:190007 <https://doi.org/10.1038/sdata.2019.7> (2019).

Response n. 14. We are grateful to the reviewer for the observation that allow us to increase the comparability of our outcomes with other studies. Comments on rare taxa processing have been added in the text (pages 19 and 20) as follows: "Rare taxa, defined as sequence variants counted less than 10 times or present in no more than 3 samples, were then removed from feces and lungs separately". For the lungs, a further manual taxa filtering criterion has been applied according to Olesen et al. (Olesen et al., Changes in Skin and Nasal Microbiome and Staphylococcal Species Following Treatment of Atopic Dermatitis with Dupilumab. *Microorganisms*. 2021 Jul

13;9(7):1487). Moreover, the following sentence has been inserted "Diversity indexes evaluation and differential analyses were all carried out on rarefied feature tables" (pages 19 and 20).

- In the method section of the MDPI paper by Olesen et al., I observed a segment detailing the manual removal of 2041 ASVs from skin data and over 300 ASVs from the nasal dataset. Additionally, they removed ASVs only classified up to the class level, those not identified as Bacteria (kingdom), and those associated with certain phyla, class, and orders, namely Cyanobacteria, Plantomycetes, Chloroflexi, Deinococcus-Thermus, etc. Based on this passage, the filtering criteria seem vague, suggesting a potentially arbitrary removal of certain ASVs, especially those challenging to interpret without negative controls. Could you elucidate your approach? Which taxa were specifically excluded? What proportion of taxa/reads got eliminated through this manual process? Was this filtering undertaken before or after rarefaction? A significant removal of ASVs like this is consequential from a statistical perspective, particularly when considering the already low number of reads you obtained for lung and fecal samples, warranting a thorough discussion in the manuscript or at least ensuring the effects are recognized and compensated for.

Response n. 17. Modified as suggested by this reviewer and reviewer n. 1. We would like to clarify that the bar plots of panels e and i of Figure 1 were deliberately not modified for the loss of clarity.

- That is great! I would suggest showing an additional supplementary figure 1e and 1i with real datapoints.

Comment n. 18. Figure 5 and Figure 7. Is the PCoA diversity analysis on Phylum or Genus or Species level? Is the Jaccard distance based on a presense/absence matrix? Was the presence/absence matrix established from rarified count data?

Response n. 18. All diversity indexes have been evaluated on tables rarefied per organ, i.e. we rarefied all feces samples and all lung samples separately and evaluated diversity indexes for all samples in each of the two rarefied tables. It follows that, within each organ, we can compare diversity indexes since they evaluate the compositional diversity with respect to the same microbiome. The PCoA diversity analysis was done on sequence variants that improved diversity resolution.

- Its unclear on how your response addresses the query about the use of a presence-absence matrix for calculating the Jaccard distance. Could you elucidate what you mean by "sequence variants that enhanced diversity resolution"? Typically, the Jaccard distance is used with binary data, such as presence/absence matrices. Was this approach adopted here? The current text doesn't seem to provide information on this aspect.

Comment n. 20. How large is the within-group variability you observed? The standard error ellipses are large and mainly overlapping. There seems to be more variance in one group than between groups. Therefore, the interpretation is not yet completely convincing.

Response n. 20. The standard error ellipses have been eliminated to better emphasize the differences shown in the box-plots and Table S3.

- Eliminating the standard error ellipses does not solve the problem of larger intra- vs. inter-group variance. Please show if this is the case.

Comment n. 21. Figure 6. How many mice per group went into the analysis? Was the data normally distributed enabling you to perform a two-way ANOVA analysis? Or what was the rationale to choose ANOVA testing from a statistical point of view? 11

Response n. 21. As indicated in the figure legend, we have used 3 control mice and between 7 to 12 mice in the treated group. The data were analyzed with the non-parametric Dunn test and not with the two-way ANOVA as erroneously written in the figure legend. We apologize for this oversight.

- The group sizes appear to be quite uneven and notably small. Can you specify the effect size for

the comparison as well as the confidence intervals for this effect size? Generally, a statistical comparison with fewer than 5 samples tends to be less meaningful. This aspect should be highlighted as a limitation in the paper.

Point-by-point reply letter

Reviewer #3 (Remarks to the Author):

I thank the authors for the comprehensive update of the method section and for offering more in-depth insights into the bioinformatics workflow. Although many of my queries have already been adequately resolved, I still have the following concerns:

Comment n. 1

Response n. 2. The figure below shows the negative results obtained upon quantification and amplification of negative controls as compared to a representative lung sample.

- *I am pleased to note that the negative controls display very low biomass. Yet, I see only one lung sample selected as representative for all the lung samples. Were the quantities consistent across lung samples, or were there instances of lung samples with lower biomass? Either way, It remains pertinent to sequence blank controls, with or without artificial spike-ins, if required. If this step was skipped, I strongly suggest acknowledging it as major limitation in your discussion.*
- *Kindly refer to Robert Dickson's publication that underscores the complexities associated with lung tissue and sequenced blank controls: Baker, J.M., Hinkle, K.J., McDonald, R.A., et al. "Whole lung tissue is the preferred sampling method for amplicon-based characterization of murine lung microbiota." *Microbiome* 9, 99 (2021). [<https://doi.org/10.1186/s40168-021-01055-4>]*

Response n. 1

We are much aware of the critical impact of contaminating DNA on results obtained from samples containing a low microbial biomass and that concurrent sequencing of negative control samples is advised. Although not done in this work, we have sequenced the negative controls in the past and confirmed the Baker's and other's finding of a distinct taxonomic composition between whole lung tissue and negative controls. Being of course unable to do the concurrent sequencing of negative control samples in the present study, we have acknowledged it as major limitation in the discussion as suggested (lines 386-389).

Comment n. 2

Response n. 11. The Qiime2 plugins used for evaluating all bioinformatic steps now have been detailed and description of parameters other than the default have been made explicit (pages 19 and 20). Line 610: In order to ease the comparison of microbial composition between groups, samples were normalized by rarefying the sequencing: feces samples were rarefied to 6700 reads, lung samples were rarefied to 3700 reads."

- Please elaborate in the manuscript on the low number of reads available after rarefaction, especially in the

light of the study by Bukin et al. (2019). Their results suggest that to achieve statistically convergent estimates of species diversity for most of the community), one should consider samples ranging from 10,000 to 15,000 reads. In such scenarios, the relative error for the diversity index estimates was less than 4%. How does this align with your outcome? See: Bukin, Y. S. et al. The effect of 16S rRNA region choice on bacterial community metabarcoding results. Sci. Data. 6:190007 (2019).

Response n. 2

Thank you for suggesting the reference of Bukin et al. (2019) that helped us to refine the details of the methods. We rarefied lung samples to 3,600 reads and thus, according to Bukin et al. (2019), each per-sample Shannon estimate could be affected by an 8% worst-case relative error. Our experiment uses, for each group of interest, the microbiome composition of at least 3 different mice and the per-group mean of each alpha index is considered for comparison purposes between groups. Considering that the relative error of the mean evaluated in each group decreases with the root-square of the number of samples in the group with respect to the starting error estimates, it follows that the estimate of alpha diversity for each sample-group (i.e. the per-group mean) shows a worst-case relative error reduced by a factor at least equal to $\sqrt{3}$, i.e. improved to less than 4.6%. This maximum worst-case error value is comparable to the 4% obtained by Bukin et al. (2019) on a single sample characterized by a sequencing depth of at least 10,000 reads. On a final note, we would like to emphasize that the scope of our work is different from that of Bukin et al. and does not have the ambition to estimate with high precision the degree of diversity in each group of mice. Instead, alpha index estimates are used to assess significance between groups, and the sample rarefaction process ensures that significances are not due to bacteria so rare that they can only be detected in groups with large numbers of reads, i.e. that the significance is not due to the greater sampling depth of the group's samples. Rarefaction of lungs to 3600 reads allows to detect taxa present with proportion greater than $1/3600=0.00027$ which is adequate for the scope of the paper. We have modified the text as follows (line 590 now line 606): "It should be noticed that sample rarefaction to 3,600 reads, while allowing the largest number of samples to be included, could lead to an 8% worst-case relative error on the per-sample alpha index estimates (Ref. 91). Since groups of interest have at least three mice, the per-group average value of alpha indexes present a worst-case relative error reduced, with respect to the per-sample estimates, by a factor at least equal to $\sqrt{3}$, i.e. improved to less than 4.6%."

Comment n. 3

Response n. 14. We are grateful to the reviewer for the observation that allow us to increase the comparability of our outcomes with other studies. Comments on rare taxa processing have been added in the text (pages 19 and 20) as follows: "Rare taxa, defined as sequence variants counted less than 10 times or present in no more than 3 samples, were then removed from feces and lungs separately". For the lungs, a further manual taxa filtering criterion has been applied according to Olesen et al. (Olesen et al., Changes in Skin and Nasal Microbiome and Staphylococcal Species Following Treatment of Atopic Dermatitis with Dupilumab. Microorganisms. 2021 Jul 13;9(7):1487). Moreover, the following sentence has been inserted "Diversity indexes evaluation and differential analyses were all carried out on rarefied feature tables" (pages 19 and 20).

- In the method section of the MDPI paper by Olesen et al., I observed a segment detailing the manual removal of 2041 ASVs from skin data and over 300 ASVs from the nasal dataset. Additionally, they removed ASVs only classified up to the class level, those not identified as Bacteria (kingdom), and those associated with certain phyla, class, and orders, namely Cyanobacteria, Plantomycetes, Chloroflexi, Deinococcus-

Thermus, etc. Based on this passage, the filtering criteria seem vague, suggesting a potentially arbitrary removal of certain ASVs, especially those challenging to interpret without negative controls. Could you elucidate your approach? Which taxa were specifically excluded? What proportion of taxa/reads got eliminated through this manual process? Was this filtering undertaken before or after rarefaction? A significant removal of ASVs like this is consequential from a statistical perspective, particularly when considering the already low number of reads you obtained for lung and fecal samples, warranting a thorough discussion in the manuscript or at least ensuring the effects are recognized and compensated for.

Response n. 3.

The removal of taxa as contaminants of respiratory microbiome was done according to the paper by Olesen et al (Ref. 89). Details of the manual filtering in the lungs, executed before rarefaction and rares filtering, have been now explicitly indicated in the paper as follows (line 561 now line 572): “thus removing taxonomies already classified as contaminants of the respiratory microbiome. i.e., ASVs classified no further than class-level, ASVs not classified as Bacteria (kingdom), and ASVs belonging to the phyla Cyanobacteria, Plantomycetes, Chloroflexi, and Deinococcus-Thermus, the class Rhodothermia, or the orders Rhizobiales, Rhodobacterales, Oceanospirillales, Azospirillales, and Rhodospirillales”. Further specified in line 563, now line 578, as follows: “After that, rarefaction strategy of samples was pursued ...”

Comment n. 4

Response n. 17. Modified as suggested by this reviewer and reviewer n. 1. We would like to clarify that the bar plots of panels e and i of Figure 1 were deliberately not modified for the loss of clarity.

- That is great! I would suggest showing an additional supplementary figure 1e and 1i with real datapoints.

Response n. 4. Done, according to the specific request in the Author checklist.

Comment n. 5

Comment n. 18. Figure 5 and Figure 7. Is the PCoA diversity analysis on Phylum or Genus or Species level? Is the Jaccard distance based on a presense/absence matrix? Was the presence/absence matrix established from rarified count data?

Response n. 18. All diversity indexes have been evaluated on tables rarefied per organ, i.e. we rarefied all feces samples and all lung samples separately and evaluated diversity indexes for all samples in each of the two rarefied tables. It follows that, within each organ, we can compare diversity indexes since they evaluate the compositional diversity with respect to the same microbiome. The PCoA diversity analysis was done on sequence variants that improved diversity resolution.

- Its unclear on how your response addresses the query about the use of a presence-absence matrix for calculating the Jaccard distance. Could you elucidate what you mean by "sequence variants that enhanced diversity resolution"? Typically, the Jaccard distance is used with binary data, such as presence/absence matrices. Was this approach adopted here? The current text doesn't seem to provide information on this aspect.

Response n. 5

We apologize for the inaccurate response. The PCoA diversity analysis was done on sequence variants that enhance diversity resolution with respect to order or species OTU clustering. Jaccard distance was calculated in Qiime2 with the 'qiime diversity beta' plugin that processes the FeatureTable[Frequency] semantic type (i.e. feature table with ASVs counts) as input, and produces the corresponding jaccard distance matrix (<https://docs.qiime2.org/2023.5/plugins/available/diversity/beta/>). We have modified the text as follows (line 600 now line 622): " with the FeatureTable[Frequency] semantic type as input."

Comment n. 6

Comment n. 20. How large is the within-group variability you observed? The standard error ellipses are large and mainly overlapping. There seems to be more variance in one group than between groups. Therefore, the interpretation is not yet completely convincing.

Response n. 20. The standard error ellipses have been eliminated to better emphasize the differences shown in the box-plots and Table S3.

- Eliminating the standard error ellipses does not solve the problem of larger intra- vs. inter-group variance. Please show if this is the case.

Response n. 6

Fully agree with you. The distributions of the first two PCoA components of the Bray-Curtis and Jaccard indices are shown in the marginal box plots of Figure 5. Due to the relatively small number of samples, nonparametric tests were used in lieu of one-way anova to assess groups significance. For each component of both beta metrics, pairwise comparisons between groups, with adjustments for multiple tests, were evaluated with the Wilcoxon rank-sum test. The results, reported in Table S3, underline that, for each dpi point, at least one PCoA component has significantly different first-order moments between the two groups, thus confirming the compositional diversity. The text has been modified as follows (line 188 now line 189): "On the contrary, for each dpi point, the two types of mice differed in compositional structure by β -diversity and ..."

Comment n. 7

Comment n. 21. Figure 6. How many mice per group went into the analysis? Was the data normally distributed enabling you to perform a two-way ANOVA analysis? Or what was the rationale to choose ANOVA testing from a statistical point of view? 11

Response n. 21. As indicated in the figure legend, we have used 3 control mice and between 7 to 12 mice in the treated group. The data were analyzed with the non-parametric Dunn test and not with the two-way ANOVA as erroneously written in the figure legend. We apologize for this oversight.

- The group sizes appear to be quite uneven and notably small. Can you specify the effect size for the comparison as well as the confidence intervals for this effect size? Generally, a statistical comparison with fewer than 5 samples tends to be less meaningful. This aspect should be highlighted as a limitation in the paper.

Response n. 7

We thank the reviewer for the suggestion that is now provided in a new paragraph of the discussion (lines 386-389).